# Response of Salt Transport and Residence Time to Geomorphologic Changes in an Estuarine System

**Wen-Cheng Liu \*, Min-Hsin Ke and Hong-Ming Liu**

Department of Civil and Disaster Prevention Engineering, National United University, Miaoli 36063, Taiwan; eric7265483@gmail.com (M.-H.K.); dslhmd@gmail.com (H.-M.L.)

**\*** Correspondence: wcliu@nuu.edu.tw; Tel.: +886-37-383357

**Abstract:** Anthropogenic changes in tidal estuaries have significantly altered bathymetry and topography over the past half century. The geomorphic-driven changes in estuarine hydrodynamics and salt transport remain unclear. To explore this issue, a SELFE (Semi-implicit Eulaerian-Lagrangian Finite Element) -based model was developed and utilized in a case study in the Danshui River, Taiwan. The model was calibrated and validated using observed water level, current, and salinity data from 2015, 2016, and 2017. The performance of the SELFE model corresponded well to the measured data. Furthermore, the validated model was utilized to analyze the hydrodynamics, residual current, limit of salt intrusion, and residence time under the predevelopment (1981) and present (2015) conditions. The predicted results revealed that the time lag of water surface elevation at both high tide and low tide under the present condition was approximately 0.5–2 h shorter under the predevelopment condition. The residual circulation under the predevelopment condition was stronger than under the present condition for low flow, causing the limit of salt intrusion to extend further upstream under the predevelopment condition compared to the limit of salt intrusion under the present condition. The calculated residence time under the predevelopment condition was longer than the residence time under the present condition. The freshwater discharge input is a dominating factor affecting the salt intrusion and residence time in a tidal estuary. A regression correlation between the maximum distance of salt intrusion and freshwater discharge and a correlation between residence time and freshwater discharge were established to predict the limit of salt intrusion and residence time under the predevelopment and present conditions with different scenarios of freshwater discharge input.

**Keywords:** hydrodynamics; salt water intrusion; residence time; three-dimensional model; geomorphologic change; estuarine system

## 1. Introduction

Estuaries, which are regarded as the transitional environment between upstream reaches that are subject to the influence of freshwater discharge and coastal oceans affected by tides, form a biodiversity-rich ecosystem. The estuarine system can be recognized as having several classifications: coastal plain estuary, bar-built estuary, delta-front estuary, fjord, rias, and others [1,2]. Material transport in tidal estuaries greatly depends on the physical hydrodynamic characteristics that form a complex environment of water levels, flows, and salinity fields. Several factors, including tidal forcing, freshwater discharge, atmospheric pressure forcing, wind stress, and channel geomorphology, determine salt transport, dissolved oxygen, nutrients, suspended sediment, biogeochemistry, and biological communities [3,4].

Most of the famous coastal cities in the world are located along estuaries that take advantage of allowing communication from rivers to overseas. For the purpose of transportation on rivers, dredged shipping channels and barrier structures have potential impacts on the hydrodynamics, residence

time, and limit of salt intrusion by altering the bathymetry and geometry that regulate the residual circulation, consequently causing changes in the estuarine ecosystem [5–9].

It is worth noting that river modifications, such as channel deepening and widening, dredging, jetty construction, tidal flat and marsh filling, and river realignment [10–17], have been carried out over the past several decades; therefore, understanding the long-term responses to salt transport, estuarine flows, and residence time in tidal estuaries as a result of anthropogenic activities is of utmost importance for further ecological management purposes, because the salt intrusion, estuarine circulation, and residence time crucially affect the water quality and sediment transport, fecal coliform, and heavy metal transport [18–22]. However, it is difficult to employ analytical analysis to better understand the impact of geomorphologic changes due to engineering activities on the hydrodynamics and residual circulation in estuaries since the complexities of topography and bathymetry, stratification and mixing, buoyancy forcing, and wind make the problem more intractable to solve. Therefore, different kinds of numerical models have become crucial tools to evaluate the responses of altering river systems, changing hydrodynamics, salt intrusion, gravitational circulation, and residence time in tidal estuaries [23–27]. For example, Velamala et al. [15] recently used a two-dimensional (2D) vertically integrated hydrodynamic model to assess the impacts of bathymetric changes on the flushing characteristics and residence time in the Amba estuary as a result of dredging carried out for navigation purposes. Andrews et al. [28] applied a three-dimensional (3D) model to quantitatively evaluate the differences in salt intrusion under two scenarios, which included the contemporary and predevelopment conditions in the San Francisco Estuary. The results indicated that salt intrusion in the predevelopment period was more sensitive to outflow than in the contemporary period. Anthropogenic modifications over the past two centuries have crucially altered tidal trapping and salt intrusion processes. Liu et al. [29] utilized an EFDC-3D (Environmental Fluid Dynamics Code-3D) to explore the changes in salt distributions in the Modaomen Estuary, China, especially during the dry season due to riverbed downcutting and large-scale land reclamation. They found that bathymetric and topographic alterations crucially increased the distance of salt intrusion and enhanced salt stratification in the estuary.

A number of publications have documented that anthropogenic modifications alter physical processes and possibly change estuarine ecosystems. Increasing the distance of salt intrusion led to changes in the ecosystem, and increasing the residence time of the water body caused more pollution to stay in the estuary for too long [15,16,23,28,29]. Therefore there is urgent need to better understanding the effects on geomorphic-driven changes on salt water intrusion and residence time in estuarine system. The primary objective of the present study is to analyze the hydrodynamics, salt intrusion, and residence time in the Danshui River estuarine system under the predevelopment (1981) and present conditions. To achieve this goal, an unstructured grid SELFE-based model was established and adopted for an estuarine system. The model was comprehensively calibrated and validated using available observational data from 2015, 2016, and 2017. Furthermore, the validated SELFE-based model was utilized to investigate the hydrodynamics, saltwater intrusion, and residence time under both the predevelopment and present conditions.

## 2. Site Description

The Danshui River estuary (Figure 1) is the largest tidal estuary in Taiwan. This estuary includes three main tributaries—the Hsintien Stream, Dahan Stream, and Keelung River. The main Danshui River begins at the confluence of the Hsintien Stream and Dahan Stream at the western boundary of Taipei City and New Taipei City and flows northward and northwestward into the Taiwan Strait. The Dahan Stream is the main tributary and has headwaters in Pintian Mountain, which is located in Hsinchu County and flows through three counties in northern Taiwan. As a river system, the main Danshui River and its three tributaries have a total length of 159 km, covering a watershed area of 2726 km$^2$ and spanning a total channel length of 82 km subjected to tidal influence [30]. The daily freshwater discharges at the Cheng-Ling Bridge, Hsiu-Lang Bridge, and Wu-Tu stations were gathered

for flow analysis. The mean river discharges are 39.0 m$^3$/s, 69.7 m$^3$/s, and 25.0 m$^3$/s, respectively, in the Dahan Stream, Hsintien Stream, and Keelung River, while the low river discharges are 2.2 m$^3$/s, 3.8 m$^3$/s, and 1.3 m$^3$/s. The average annual flow rate and surface area are 6.6 × 10$^9$ m$^3$ and 1.74 × 10$^9$ m$^2$ approximately, respectively [31]. The positions of the tidal limit are located at the Cheng-Ling Bridge in Dahan Stream, the Hsiu-Lang Bridge in Hsintien Stream, and the Jiang-Bei Bridge in the Keelung River during low/normal flow conditions (see Figure 1).

According to the analysis of observational data at tidal gauges, the semidiurnal tide (M$_2$) is the principal tidal component at the river mouth of the Danhsui River, with a mean tidal range of 2.39 m. During the normal flow condition, the water volume between the highest tide level and the lowest tide level called the tidal prism is about 6.5 × 10$^7$ m$^3$ [32].

According to the literature [30], the phase difference between the water surface elevation and tidal velocity in time series displays a standing wave characteristic indicating that the tidal amplitude and velocity exhibit 90° phase discrepancy. The standing wave affects the sediment and heavy metal transport in tidal estuaries. No phase difference was found at the upstream reaches where were beyond the tidal imit. Chen and Liu [32] demonstrated that a marked tidal asymmetry occurred in the Danshui River estuary.

Salinity in tidal estuaries varies with the intra-tidal time scale as a result of ebb and flood flows and freshwater discharge inflow. Liu [33] found that classical two-layer estuarine circulation occurred in the Danshui River estuary. As the freshwater discharge decreases, the two-layer circulation is increased. The distance of salt intrusion reaches approximately 25 km in the Dahan Stream from the river mouth of the Danshui River, approximately 4 km in the Hsintien Stream from its stream mouth, and approximately 12 km in the Keelung River from its confluence with the Danshui River during low-flow conditions. Residence time represents a vital index for assessing the material transport (i.e., neutrally buoyant material) of estuaries. Generally, this index is adopted to evaluate the removal rate of estuarine pollutants transported by river discharge. During the normal weather condition, the influence of waves at the mouth is not important. Therefore the wave action is not included in the model simulations. The sediment transport of the Danshui River comes from the upstream watershed. The annual sediment yield in the Danshui River is about 1.145 × 10$^6$ t [34].

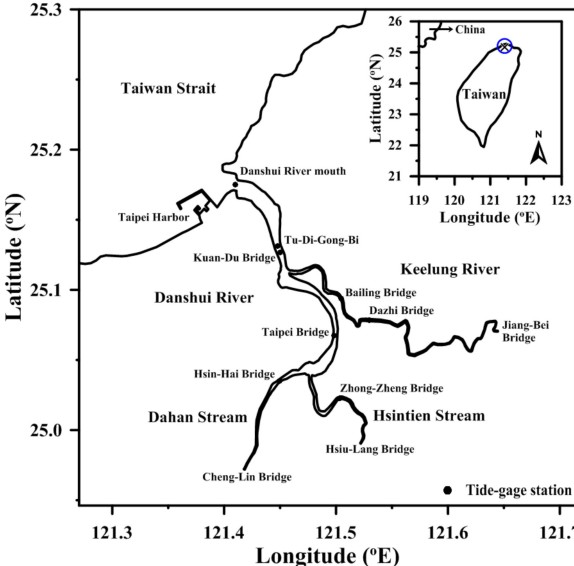

**Figure 1.** Map of the Danshui River estuarine system in northern Taiwan.

Three important engineering facilities were built after 1980 in the Danshui River system. (1) The Feitsui Reservoir is located downstream of Beishi Creek, a tributary of the Hsintien Stream, and is 30 km from the Taipei metropolitan area. The reservoir has an initial design capacity of 406 × 10$^6$ m$^3$

constructed at the upper reaches of the Hsintien Stream. The reservoir was completed in 1987 to supply domestic water to residents who live in metropolitan Taipei and New Taipei City. (2) The channel realignment in the Keelung River began in 1991 and was competed in 1993 to shortcut two meanders for flood mitigation and to resolve the housing problems of more than 13,000 residents living in the floodplains. (3) The Yuansantze Flood Diversion Works Project, constructed in the upper river reach of the Keelung River, began in 2001 and was completed in 2004 to mitigate inundation disasters and to reduce economic losses. These engineering facilities have altered the topography and bathymetry in the channels of the Danshui River system.

To obtain a better understanding of the geomorphologic changes in the Danshui River estuarine system, bathymetric surveys conducted by the Taiwan Water Resources Agency (TWRA) in 1980 and 2015 were utilized for further analysis and investigation. Each cross section in the channel at approximately 0.5 km intervals was surveyed. The results revealed that there were many bathymetric and topographic changes in the river system. A comparison of the riverbed profiles between 1981 and 2015 along the river axis in the main streams and three tributaries is shown in Figure 2. The maximum change occurred at a distance of 10 km from the river mouth in the Danshui River-Dahan Stream, where the depth changed from 13 m in 1981 to 4.89 m in 2015 (Figure 2a). The maximum change in the Hsintien Stream was found at a distance of 3 km from the mouth of the Hsintien Stream, where the depth altered from 9 m in 1981 to 3.87 m in 2015. Deposition was found in the lower reaches of the Hsintien Stream (Figure 2b). In the Keelung River, the maximum change in the riverbed is located 8 km from the mouth of the Keelung River, where the depth increased from 7 m in 1981 to 11.53 m in 2015. Obviously, a scouring phenomenon occurred downstream in the Keelung River (Figure 2c).

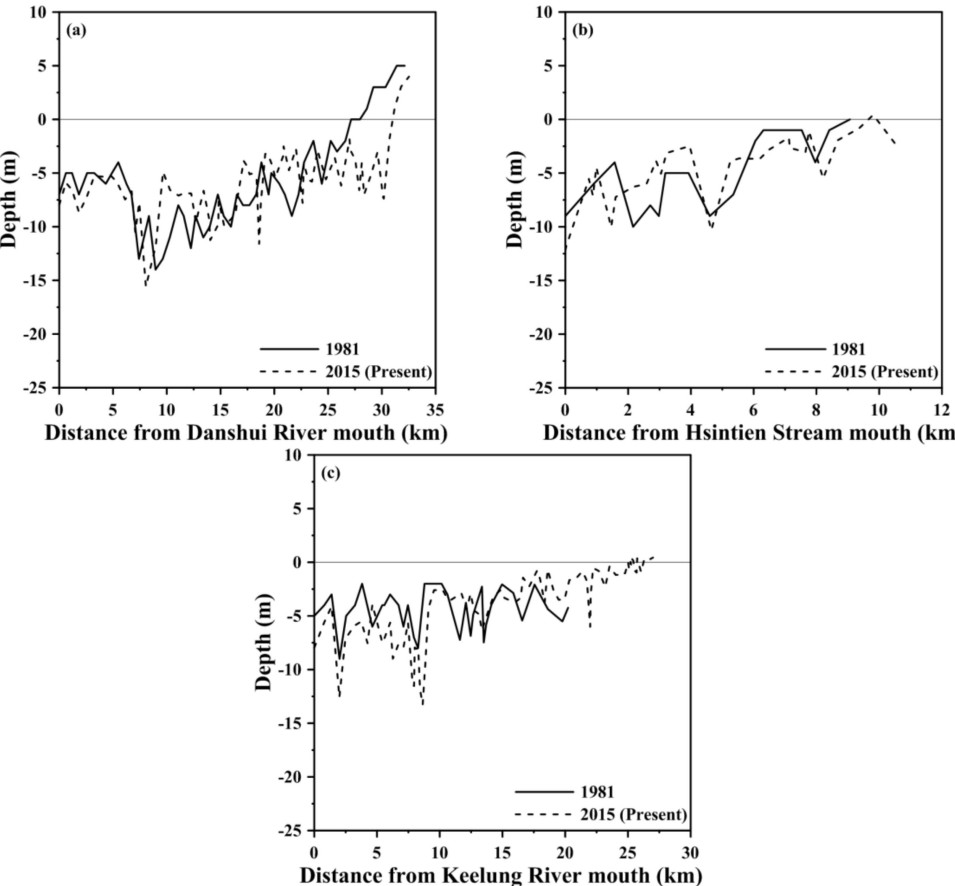

**Figure 2.** Comparison of the bottom profiles in 1981 and 2015 in the (**a**) Danshui River–Dahan Stream, (**b**) Hsintien Stream, and (**c**) Keelung River.

A comparison of 1981 and 2015 with regard to the average width below the mean sea level in the main river and three tributaries is shown in Figure 3. The maximum average width increased by 181 m, 173 m, and 220 m in the Danshui River-Dahan Stream, Hsintien Stream, and Keelung River, respectively. This comparison showed an extensive widening of the channel downstream of the Hsintien Stream and the Keelung River (Figure 3b,c). However, the average width did not change drastically in the lower reaches of the Danshui River–Dahan Stream (Figure 3a).

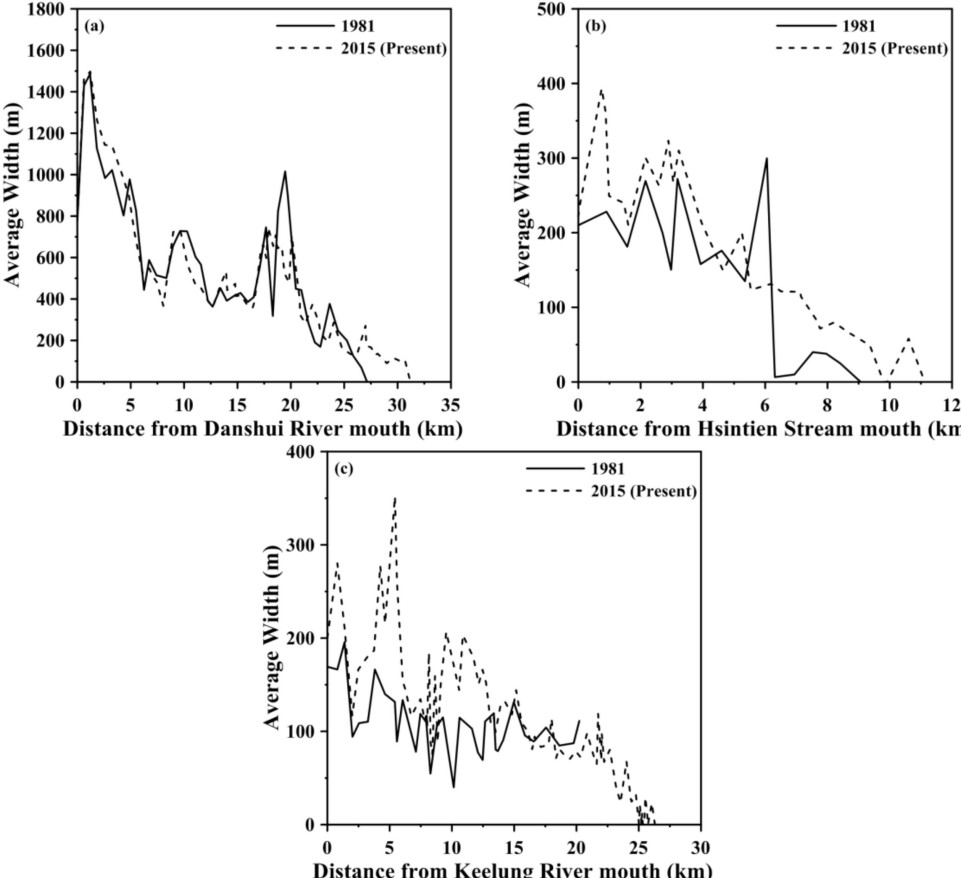

**Figure 3.** Comparison of the averaged width below mean sea level in 1981 and 2015 in the (**a**) Danshui River–Dahan Stream, (**b**) Hsintien Stream, and (**c**) Keelung River.

## 3. Methods

A number of models have been developed and applied to extensively understand the hydrodynamic characteristics in tidal estuaries employing one-dimensional (1D), two-dimensional (2D), or three-dimensional (3D) frameworks. Recently, high-resolution, unstructured grid 3D models have increasingly been utilized to resolve the complex topographic and bathymetric features and sharp gradients between water masses in tidal estuaries and coasts. In this study, using the same 3D hydrodynamic engine, two models were built to investigate the Danshui River estuarine system under the predevelopment condition and under the present condition.

### 3.1. Hydrodynamic Model

The hydrodynamics were simulated using an unstructured grid, the Semi-implicit Eulaerian-Lagrangian Finite Eement model called SELFE [35]. This model solves 3D shallow water equations and transport process for conservative substances such as salt. The hydrostatic assumption and Boussinesq approximation are adopted to build the shallow water equations. The free-surface elevation, 3D velocity, and salinity in SEFLE are principal variables to be solved in Cartesian coordinates.

SELFE utilizes triangular grids in the horizontal plane that are inserted in the vertical direction to construct a 3D prismatic mesh. To discretize the free-surface elevation and velocity in the horizontal direction, algorithms with nonconforming and linear continuous functions are used. In the vertical direction, the terrain following the S layer is arranged on the top, followed by the equipotential z layer [36].

An implicit free-surface equation combined with the algorithm for wet and dry grids was carried out in the SELFE-based model. The vertical diffusion and surface-surface are treated with an implicit method. An Eulerian-Lagrangian approach is adopted to solve the advection of momentum over time. This formation forms the restrictive constraint with the Courant number to allow time steps as long as possible without the instability problem in the computational process. An explicit mass conservative upwind scheme is used to solve the salt transport equation. Therefore, a shorter time step is needed to retain the conservation and monotonicity properties. The element size, local velocity, and salt fields are the primary restrictions to choosing the time step to run the model.

The General Ocean Turbulence Model [8,37], supplying turbulent eddy viscosity and diffusivity in the vertical direction, is used to calculate the vertical subgrid-scale mixing. A turbulence closure model ($k - \varepsilon$ model) with the Canuto-A stability function [38,39], which is embedded in the General Ocean Turbulence Model, is utilized in this study.

### 3.2. Grid Establishment for Computation

The modeling domain was extended from the estuarine system to the continental shelf (Figure 4). To establish the meshes for running the model, bathymetric and topographic data measured in 1981 and 2015 from the estuarine system and continental shelf were gathered from government agencies in Taiwan. A field survey was conducted by the TWRA to measure the cross-sectional profiles every 300–500 m along the river. The resolution of 10 km for bathymetric and topographic data in the coastal sea was measured by the National Center for Ocean Research. Two kinds of meshes for the predevelopment condition (1981) and present condition (2015) were generated for model simulation. To save computational time, the fine-grid and coarse-grid resolutions were yielded in the estuarine system and continental shelf, respectively.

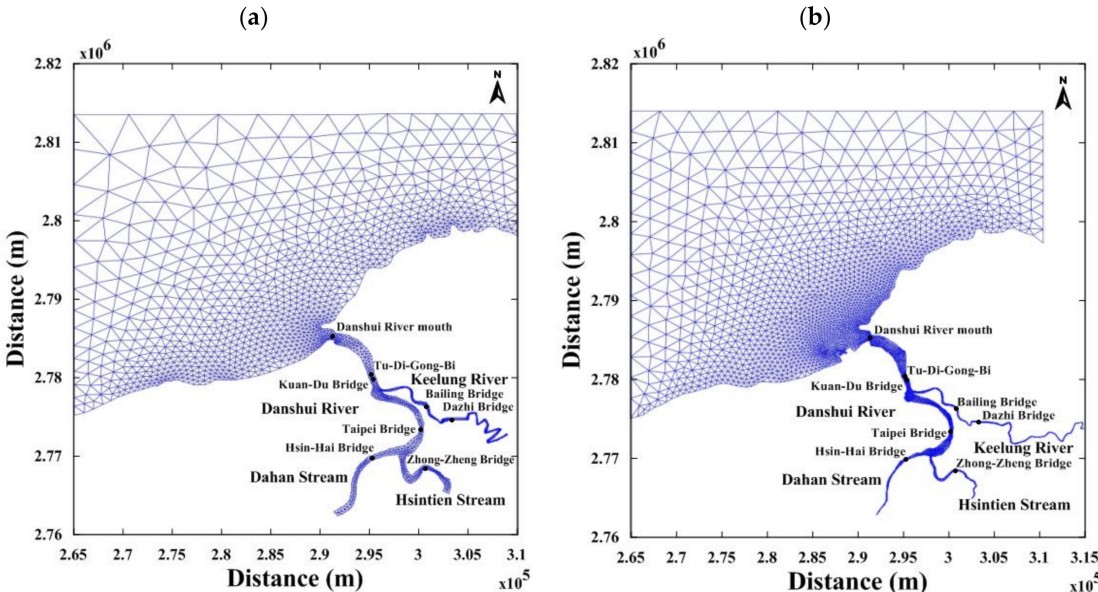

**Figure 4.** Unstructured grid in the horizontal plane for model simulation under (**a**) the predevelopment condition (1981) and (**b**) the present condition (2015). Note that two meanders in the Keelung River were shortcut in 1994 for flood mitigation.

A total of 2354 elements/3854 nodes were set up in the horizontal domain, where the horizontal mesh resolution ranges from 75 m in the estuarine system to 5500 m in the continental shelf under the predevelopment condition (Figure 4a). A total of 6335 elements/4092 nodes were generated in the horizontal plane. The horizontal mesh resolution ranges from 50 m in the estuarine system to 3200 m in the continental shelf under the present condition (Figure 4b). Hybrid layers are adopted in the vertical direction, 10 layers in the S-coordinates, and 10 layers in the Z-coordinates. The computational time step, 120 s, was utilized in the model simulation without the problem of numerical instability.

### 3.3. Boundary and Initial Conditions

To search for the upstream boundary that is not subjected to the tidal effect, the modeling domain is extended to the limit of the tide in the three tributaries. The locations at the Cheng-Lin Bridge (Dahah Stream), Hsiu-Lang Bridge (Hsintien Stream), and Jiang-Bei Bridge (Keelung River) were designated as upstream boundaries where freshwater discharge was specified. The time-series water level was generated by five tidal components (i.e., $M_2$, $S_2$, $N_2$, $K_1$, and $O_1$) specified at the open boundaries to drive the model run. Five tidal components with resolution of 5 km in coastal sea were obtained from the National Center for Ocean Research.

The salinity concentrations at the open boundaries and upstream boundaries were specified using 35 ppt (parts per thousand) and 0 ppt, respectively. The initial conditions adopted were mean values of 1.5 m, 0.6 m/s, and 25 ppt for the water level, velocity, and salinity, respectively, utilized to warm up the model. We also tested the rest condition (i.e., water level 0 m and velocity 0 m/s) and the climatological salinity field used as initial conditions. However, the 30-day warm up time was needed to reach an equilibrium state before the model results were compared with the observations.

### 3.4. Criteria for Model Performance

Four criteria were used to quantify the ability of the model based on a comparison of the modeling results and observational data to characterize the SELFE-based model in estuarine systems. These criteria include the mean absolute error (MAE), root mean square error (RMSE), correlation coefficient ($r$), and skill index of agreement. The skill index represents a nondimensional score that was previously developed by Wilmott [40] and, in recent years, has been widely used in the literature [4,26,41,42].

The criteria for the correlation coefficient ($r$) and *Skill* are defined using Equations (1) and (2):

$$r = \frac{\sum\limits_{i=1}^{n}(M_i - \bar{M})(O_i - \bar{O})}{\sqrt{\sum\limits_{i=1}^{n}(M_i - \bar{M})^2}\sqrt{\sum\limits_{i=1}^{n}(O_i - \bar{O})^2}} \tag{1}$$

$$Skill = 1 - \frac{\sum\limits_{i=1}^{n}|M_i - O_i|^2}{\sum\limits_{i=1}^{n}\left[\left|M_i - \bar{O}\right| + \left|O_i - \bar{O}\right|\right]^2} \tag{2}$$

where $n$ denotes the total number, $M$ represents the modeling value, $O$ expresses the observed value, $\bar{M}$ indicates the mean value of model prediction ($= \frac{1}{n}\sum\limits_{i=1}^{n}M_i$), and $\bar{O}_o$ denotes the mean value of observation ($= \frac{1}{n}\sum\limits_{i=1}^{n}O_i$).

The skill values of 1.0 and 0.0 indicate perfect and poor model performances, respectively. To clearly distinguish the model performance for predictive skill, Chen et al. [43] classified that a skill value of 0.65–1.0 represents excellent model performance. The values of 0.5–0.65 and 0.2–0.5 denote very good and good model performances, respectively.

## 4. Model Calibration and Validation

To confirm the availability and capability of the SELFE model, observational data were utilized to perform model calibration and validation. The observed data of the water surface elevation, velocity, and salinity in 2015, 2016, and 2017 were yielded from the TWRA for calibrating and validating the hydrodynamic model. The hourly water surface elevation was observed at the Danshui River mouth, Tu-Di-Gong-Bi, Taipei Bridge, Hsin-Hai Bridge, Zhong-Zheng Bridge, Bailing Bridge, and Dazhi Bridge gauge stations. Every half-hour longitudinal velocity was measured at the Kuan-Du Bridge, Taipei Bridge, Hsin-Hai Bridge, Zhong-Zheng Bridge, and Bailing Bridge, while half-hourly salinity was measured at the Kuan-Du Bridge, Taipei Bridge, and Bailing Bridge.

### 4.1. Model Calibration

The measured data from 2015 and 2016 were used for model calibration in the first step. Bottom friction in the SELFE model can affect the water surface elevation and current since bottom friction scatters to eliminate tidal wave energy. The daily freshwater discharges at the upstream boundaries of the three tributaries were input into the hydrodynamic model. The computed time-series water surface elevations and measured data during 27 June–3 July 2015, at seven gauge stations were compared (Figure 5). The water surface elevation was predicted well at the Danshui River mouth but slightly underestimated during high tide at other stations. In general, the simulation results reproduced the variations in water surface elevation very well. The underestimated water surface elevations during high tide would be the reason that the daily freshwater discharges were used as upstream boundaries in three major tributaries. The MAEs at seven gauge stations, the Danshui River mouth, Tu-Di-Gong-Bi, Taipei Bridge, Hsin-Hai Bridge, Zhong-Zheng Bridge, Bailing Bridge, and Da-Zhi Bridge, were 0.041 m, 0.263 m, 0.106 m, 0.258 m, 0.188 m, 0.142 m, and 0.142 m, respectively, while the RMSEs were 0.051 m, 0.323 m, 0.130 m, 0.316 m, 0.219 m, 0.164 m, and 0.178 m, respectively. The $r$ and skill values exceeded 0.95 for all the gauge stations (see Table 1). Table 1 also presents the model performance of water surface elevation during 1–7 July 2016 for model calibration.

**Table 1.** Statistical errors between measured and simulated water surface elevations.

| Date (Period) | Error Index | Model Calibration | | | | | | |
|---|---|---|---|---|---|---|---|---|
| | | Danshui River Mouth | Tu-Di-Gong-Bi | Taipei Bridge | Hsin-Hai Bridge | Zhong-Zheng Bridge | Bailing Bridge | Dazhi Bridge |
| 27 June ~ 3 July 2015 | MAE (m) | 0.041 | 0.263 | 0.106 | 0.258 | 0.188 | 0.142 | 0.142 |
| | RMSE (m) | 0.051 | 0.323 | 0.130 | 0.316 | 0.219 | 0.164 | 0.178 |
| | *r* | 0.998 | 0.952 | 0.994 | 0.956 | 0.987 | 0.995 | 0.982 |
| | *Skill* | 0.999 | 0.960 | 0.994 | 0.964 | 0.983 | 0.989 | 0.985 |
| 1 ~ 7 July 2016 | MAE (m) | 0.052 | 0.153 | 0.078 | 0.440 | 0.435 | 0.134 | 0.172 |
| | RMSE (m) | 0.063 | 0.235 | 0.097 | 0.596 | 0.655 | 0.163 | 0.211 |
| | *r* | 0.999 | 0.970 | 0.997 | 0.927 | 0.877 | 0.985 | 0.970 |
| | *Skill* | 0.999 | 0.985 | 0.998 | 0.883 | 0.830 | 0.991 | 0.984 |
| 21 ~ 27 June 2017 | MAE (m) | 0.081 | 0.240 | 0.111 | 0.367 | 0.343 | 0.195 | 0.142 |
| | RMSE (m) | 0.096 | 0.307 | 0.136 | 0.443 | 0.425 | 0.219 | 0.190 |
| | *r* | 0.998 | 0.968 | 0.995 | 0.953 | 0.961 | 0.995 | 0.985 |
| | *Skill* | 0.998 | 0.975 | 0.996 | 0.946 | 0.946 | 0.986 | 0.988 |

Figure 6 shows the comparison between the measured data for the depth-averaged velocity in time-series along the channel and the computed velocities at five measurement stations on 30 June 2015. The model results showed that the current at flood tide was weaker and denoted a shorter duration

than the current at ebb tide. This shorter duration of the current was the reason that the interaction between the tide propagating from the continental shelf and the freshwater discharge flowing from upstream reaches resulted in tidal asymmetry. The MAEs at five stations, Kuan-Du Bridge, Taipei Bridge, Hsin-Hai Bridge, Zhong-Zheng Bridge, and Da-Zhi Bridge, were 0.113 m/s, 0.188 m/s, 0.082 m/s, 0.110 m/s, and 0.193 m/s, respectively, while the RMSEs were 0.159 m/s, 0.211 m/s, 0.098 m/s, 0.136 m/s, and 0.208 m/s, respectively. The *r* and skill values ranged from 0.822 ~ 0.95 and 0.672 ~ 0.926, respectively, indicating that the model performance was excellent (Table 2). This table also shows the statistical metrics between the computed and measured tidal velocity on 4 July 2106 for model calibration. Based on the model calibration results, a constant value of roughness height ($z_0 = 0.004$ m) was adopted in the model simulation.

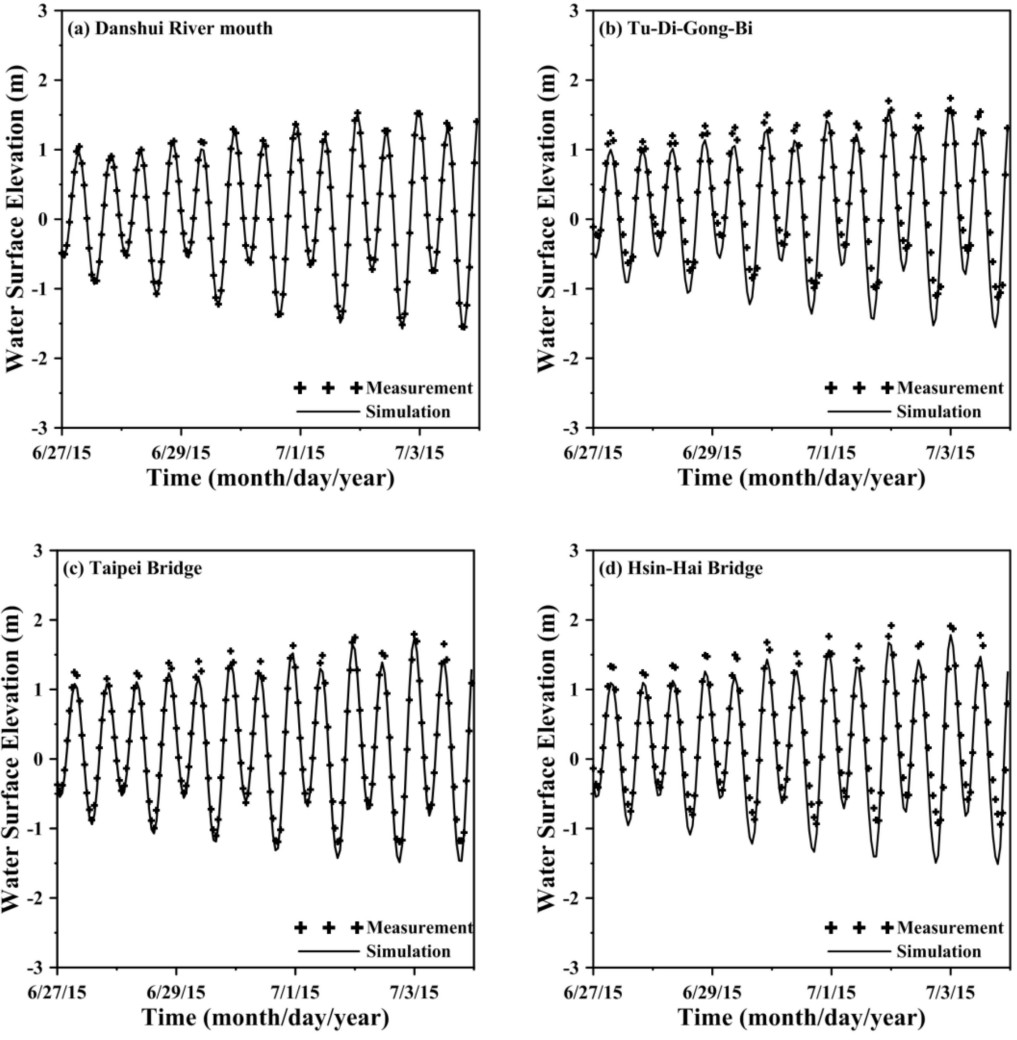

**Figure 5.** *Cont.*

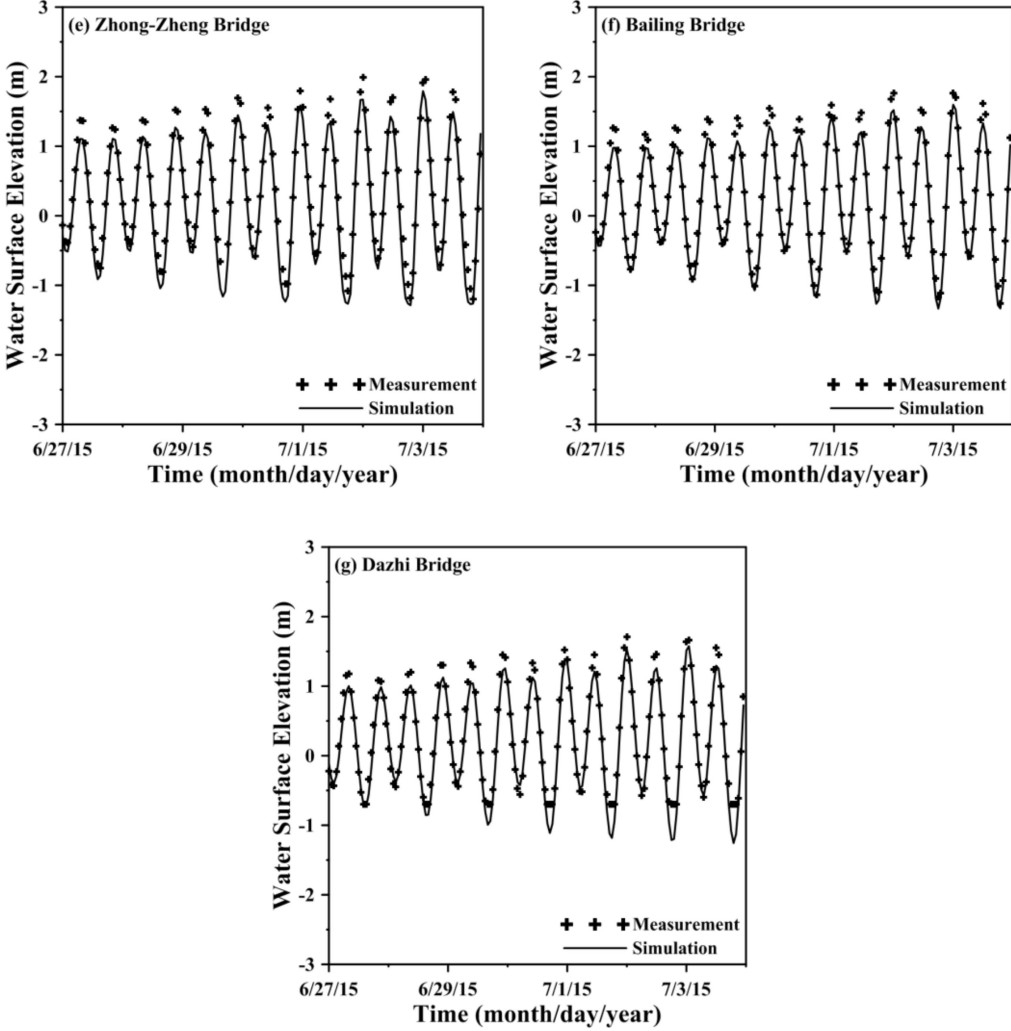

**Figure 5.** Comparison of the simulated and measured water surface elevations during the period 27 June to 3 July 2015, at the (**a**) Danshui River mouth, (**b**) Tu-Di-Cong-Bi, (**c**) Taipei Bridge, (**d**) Hain-Hai Bridge, (**e**) Zhong-Zheng Bridge, (**f**) Bailing Bridge, and (**g**) Dazhi Bridge (model calibration).

**Table 2.** Statistical errors between measured and simulated tidal velocities.

| Date | Error Index | Model Calibration | | | | |
|---|---|---|---|---|---|---|
| | | Kuan-Du Bridge | Taipei Bridge | Hsin-Hai Bridge | Zhong-Zheng Bridge | Bailing Bridge |
| 30 June 2015 | MAE (m/s) | 0.113 | 0.188 | 0.082 | 0.110 | 0.193 |
| | RMSE (m/s) | 0.159 | 0.211 | 0.098 | 0.136 | 0.208 |
| | *r* | 0.950 | 0.922 | 0.822 | 0.859 | 0.867 |
| | *Skill* | 0.926 | 0.917 | 0.901 | 0.672 | 0.827 |
| 4 July 2016 | MAE (m/s) | 0.128 | 0.302 | 0.232 | 0.239 | 0.337 |
| | RMSE (m/s) | 0.162 | 0.331 | 0.262 | 0.260 | 0.383 |
| | *r* | 0.961 | 0.898 | 0.764 | 0.703 | 0.965 |
| | *Skill* | 0.957 | 0.828 | 0.642 | 0.501 | 0.742 |
| 24 June 2017 | MAE (m/s) | 0.398 | 0.347 | 0.189 | 0.293 | 0.385 |
| | RMSE (m/s) | 0.450 | 0.426 | 0.223 | 0.304 | 0.444 |
| | *r* | 0.950 | 0.613 | 0.755 | 0.429 | 0.980 |
| | *Skill* | 0.849 | 0.739 | 0.769 | 0.341 | 0.741 |

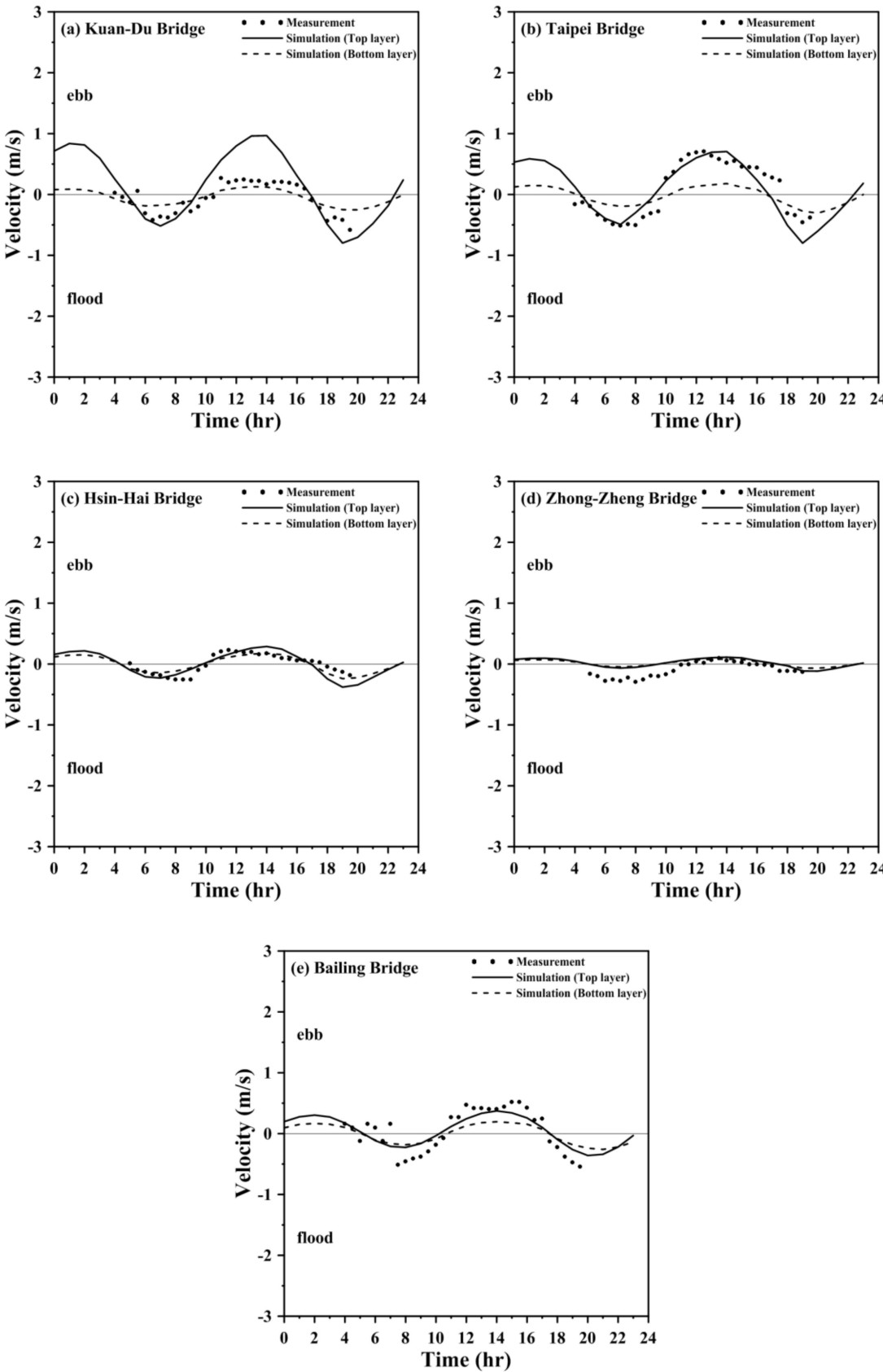

**Figure 6.** Comparison of the simulated longitudinal velocity with time-series data on 30 June 2015, at the (**a**) Kuan-Du Bridge, (**b**) Taipei Bridge, (**c**) Hsin-Hai Bridge, (**d**) Zhong-Zheng Bridge, and (**e**) Bailing Bridge (model calibration).

In tidal estuaries, salinity concentration usually acts as the natural tracer utilized for calibrating turbulent mixing processes. The spatial and temporal variations in salinity are greatly influenced by several driving factors, including tidal action, velocity, river discharge, residual circulation induced by density differences, and mixing mechanisms. The model was calibrated with the time-series salinity measured at the Tu-Di-Gong-Bi, Taipei Bridge, and Bailing Bridge. A comparison of the measured and simulated time-series salinities on 30 June 2015, is shown in Figure 7. The computed and measured salinities with averaged salinity plus/minus a standard deviation at both the surface and bottom layers are displayed in this figure. The computed salinity reproduced well the measured salinity at both ebb tide and flood tide. The MAEs ranged from 0.498–4.544 ppt among the three stations, while the RMSEs ranged from 0.648–4.995 ppt. The *r* and skill scores were 0.931–0.968 and 0.842–0.955, respectively, indicating that the model performance was excellent. Table 3 also presents the statistical errors between the simulated and measured salinities on 4 July 2106, for model calibration.

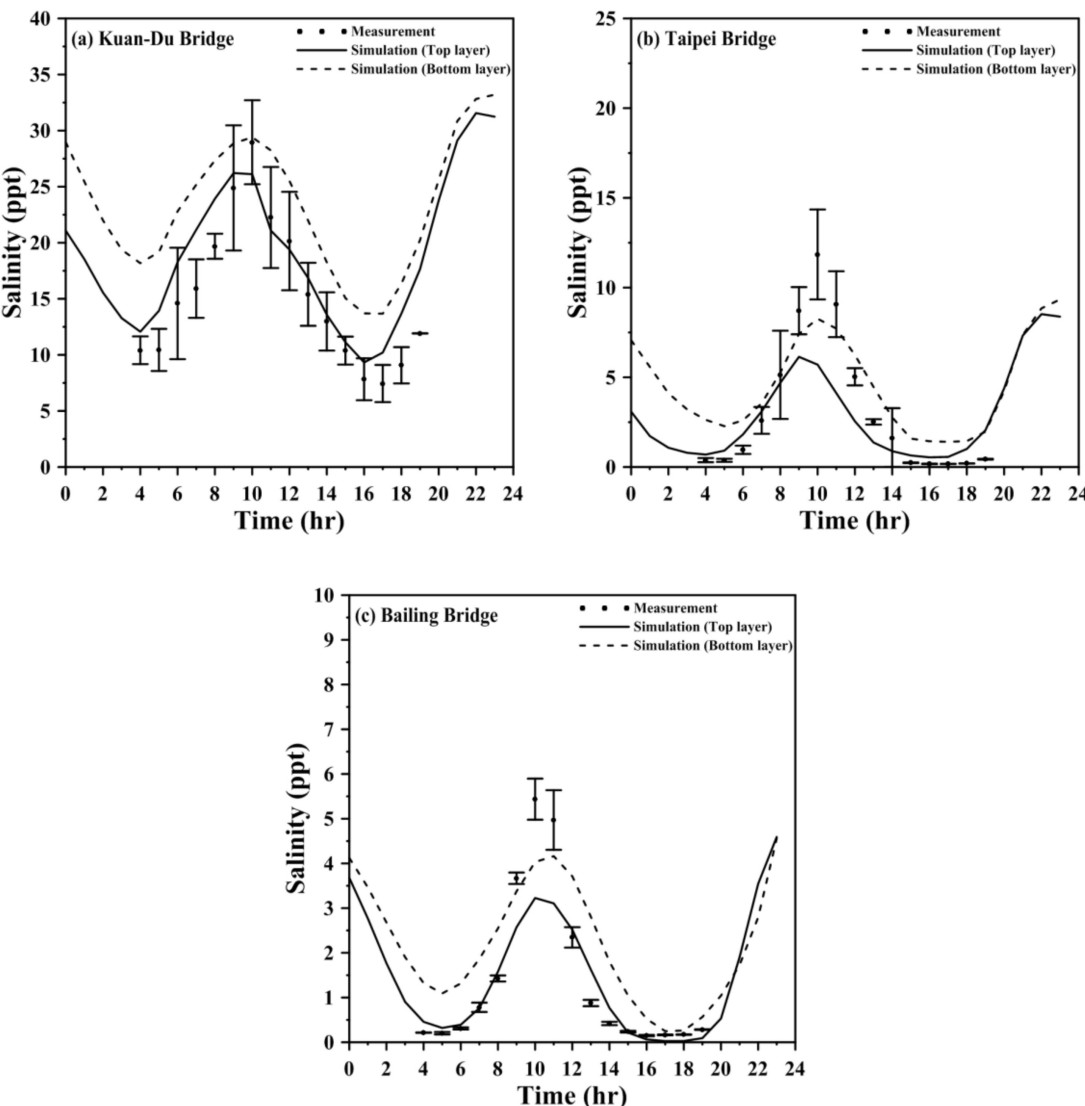

**Figure 7.** Comparison of the simulated salinity with time-series data on 30 June 2015, at the (**a**) Kuan-Du Bridge, (**b**) Taipei Bridge, and (**c**) Bailing Bridge (model calibration).

**Table 3.** Statistical errors between measured and simulated salinities.

| Date | Error Index | Model Calibration | | |
|---|---|---|---|---|
| | | **Kuan-Du Bridge** | **Taipei Bridge** | **Bailing Bridge** |
| 30 June 2015 | MAE (ppt) | 4.544 | 1.238 | 0.498 |
| | RMSE (ppt) | 4.995 | 1.655 | 0.648 |
| | *r* | 0.931 | 0.968 | 0.942 |
| | *Skill* | 0.842 | 0.918 | 0.955 |
| 4 July 2016 | MAE (ppt) | 5.508 | 0.155 | 0.251 |
| | RMSE (ppt) | 6.404 | 0.287 | 0.326 |
| | *r* | 0.885 | 0.883 | 0.939 |
| | *Skill* | 0.825 | 0.836 | 0.812 |
| 24 June 2017 | MAE (ppt) | 5.942 | 0.425 | 0.148 |
| | RMSE (ppt) | 6.414 | 0.691 | 0.157 |
| | *r* | 0.928 | 0.899 | 0.974 |
| | *Skill* | 0.880 | 0.860 | 0.881 |

*4.2. Model Validation*

After model calibration, the other dataset was used for model validation. In this study, the measured data from 2017 were adopted to compare the simulation results utilizing the well-calibrated SELFE-based model.

The modeling water surface elevations in the time series and measured data during 21–27 June 2107 at seven gauge stations were compared for model validation (Figure 8). The water surface elevation was reproduced well at the Danshui River mouth and Tu-Di-Gong-Bi, but the computed water surface elevation slightly underestimated the measured data during high tide at the other stations. The MAEs at the seven stations were 0.081 m, 0.240 m, 0.111 m, 0.367 m, 0.343 m, 0.195 m, and 0.142 m, respectively, while the RMSEs were 0.096 m, 0.307 m, 0.136 m, 0.443 m, 0.425 m, 0.219 m, and 0.190 m, respectively. The model performance with the MAE and RMSE values for model validation was slightly worse than that for model calibration. The *r* and skill values exceeded 0.94 for all the gauge stations (see Table 1 for model validation). The model performance for water surface elevation was regarded as excellent.

The comparison between the measured velocity in the time series along the channel and the computed velocities at five stations on 24 June 2017 is presented in Figure 9. The model results agreed with the measured velocities, except for the Zhong-Zheng Bridge and Bailing Bridge, where the computed velocities underestimated the measured results during ebb tide (Figure 9d,e). It is the reason that the Zhong-Zheng Bridge is close to the upstream boundary and is not subject to tidal effect. The interpolation of daily freshwater discharge at upstream boundary cannot reflect the hourly flow. The underestimated measured results during ebb tide at the Bailing Bridge would be the reason that the roughness height used at this station was not suitable. The *r* and skill values ranged from 0.429–0.98 and 0.341–0.849, respectively, showing that the lowest *r* and skill values were found at the Zhong-Zheng Bridge (Table 2). The model performance for tidal velocity was in the range of good to excellent.

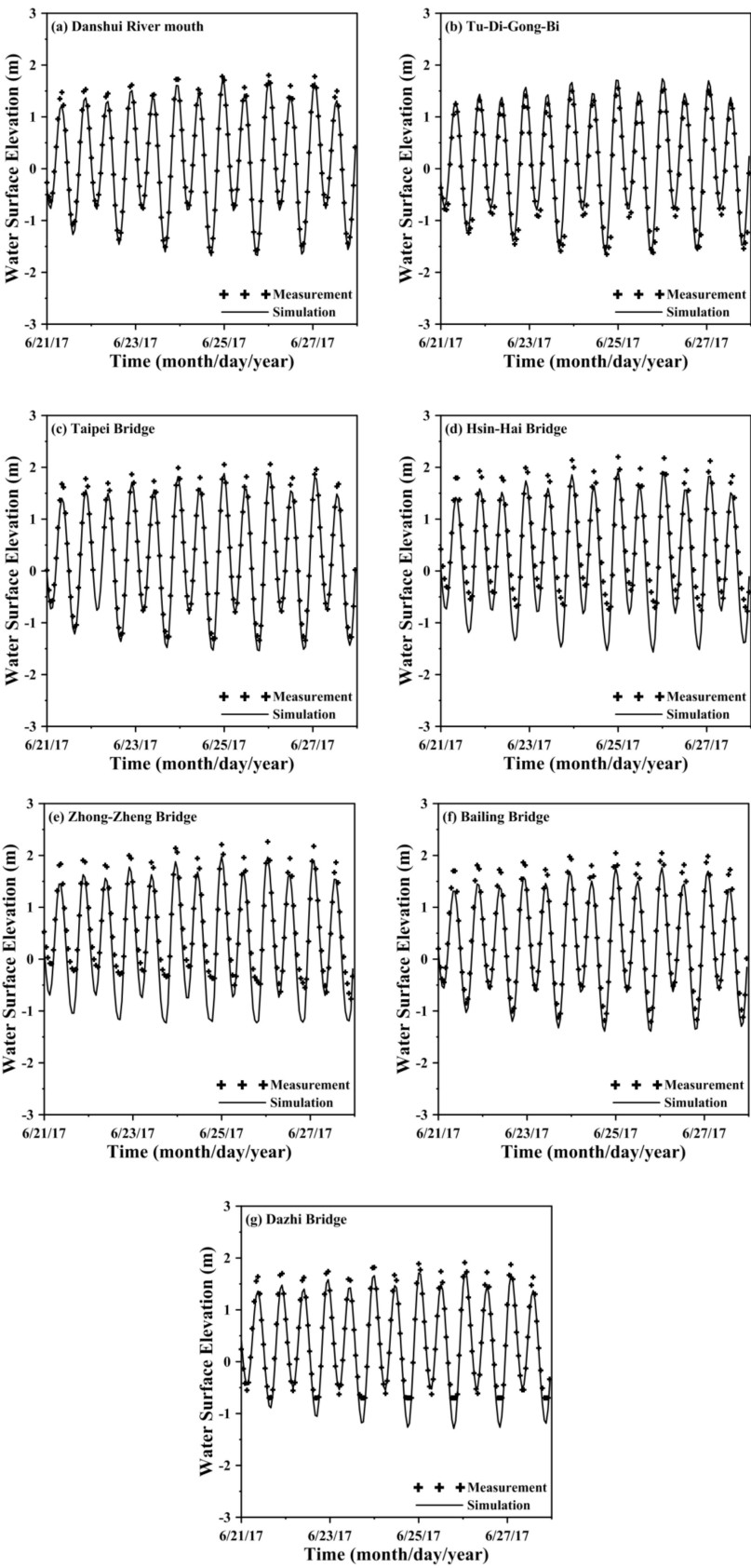

**Figure 8.** Comparison of the simulated and measured water surface elevations during the period of 21 June to 27 June 2017 at the (**a**) Danshui River mouth, (**b**) Tu-Di-Cong-Bi, (**c**) Taipei Bridge, (**d**) Hain-Hai Bridge, (**e**) Zhong-Zheng Bridge, (**f**) Bailing Bridge, and (**g**) Dazhi Bridge (model validation).

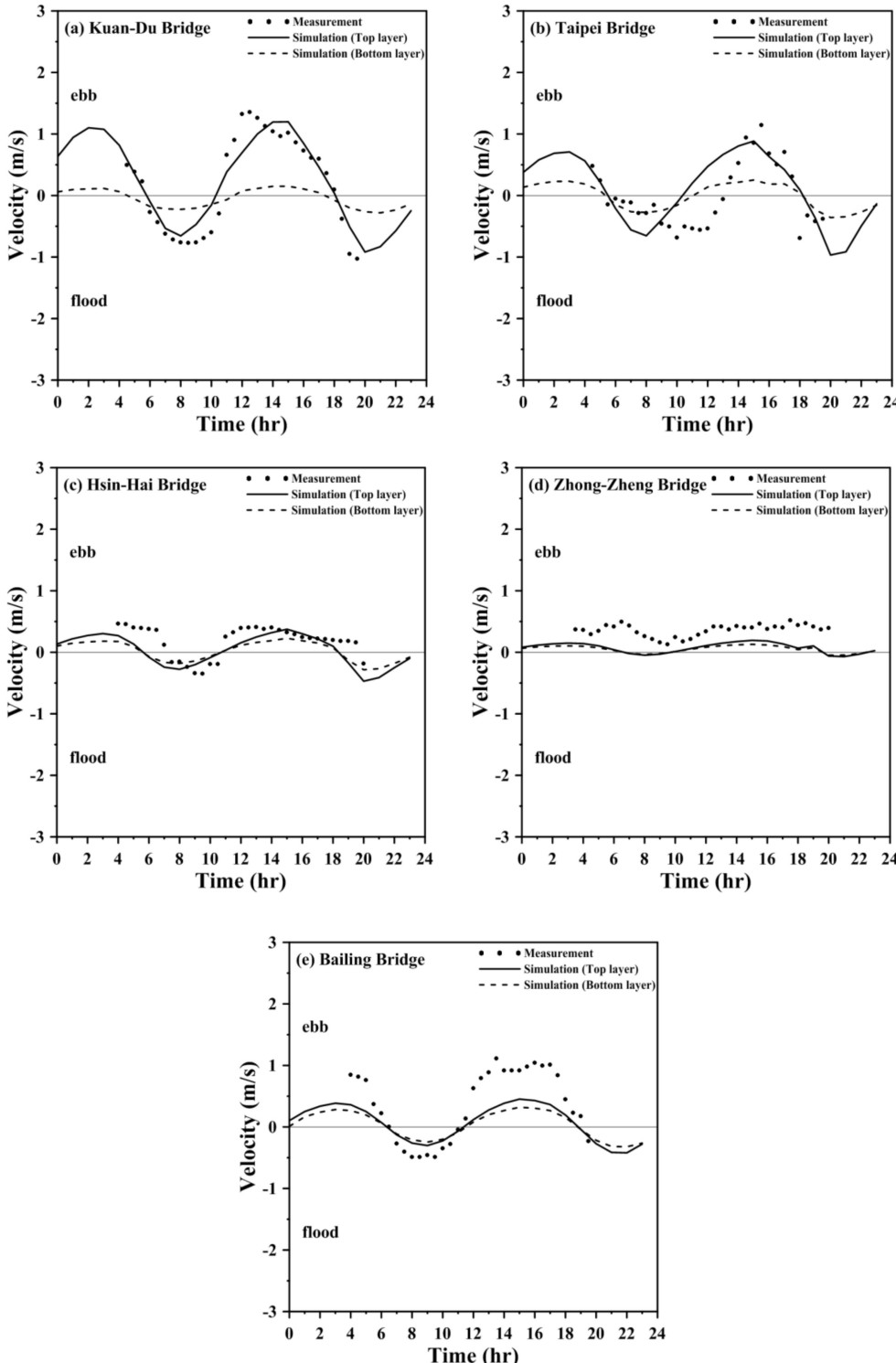

**Figure 9.** Comparison of the simulated longitudinal velocity with time-series data on 24 June 2017 at the (**a**) Kuan-Du Bridge, (**b**) Taipei Bridge, (**c**) Hsin-Hai Bridge, (**d**) Zhong-Zheng Bridge, and (**e**) Bailing Bridge (model validation).

The comparison of the measured and simulated time-series salinities on 24 June 2017 are shown in Figure 10. The model overestimated the measured salinity during ebb tide at the Kuan-Du Bridge (Figure 10a) but satisfactorily predicted the salinity at the Taipei Bridge and Bailing Bridge. However,

the *r* and skill values ranged from 0.899–0.974 and 0.86–0.881, respectively, indicating that the model performance for salinity was excellent.

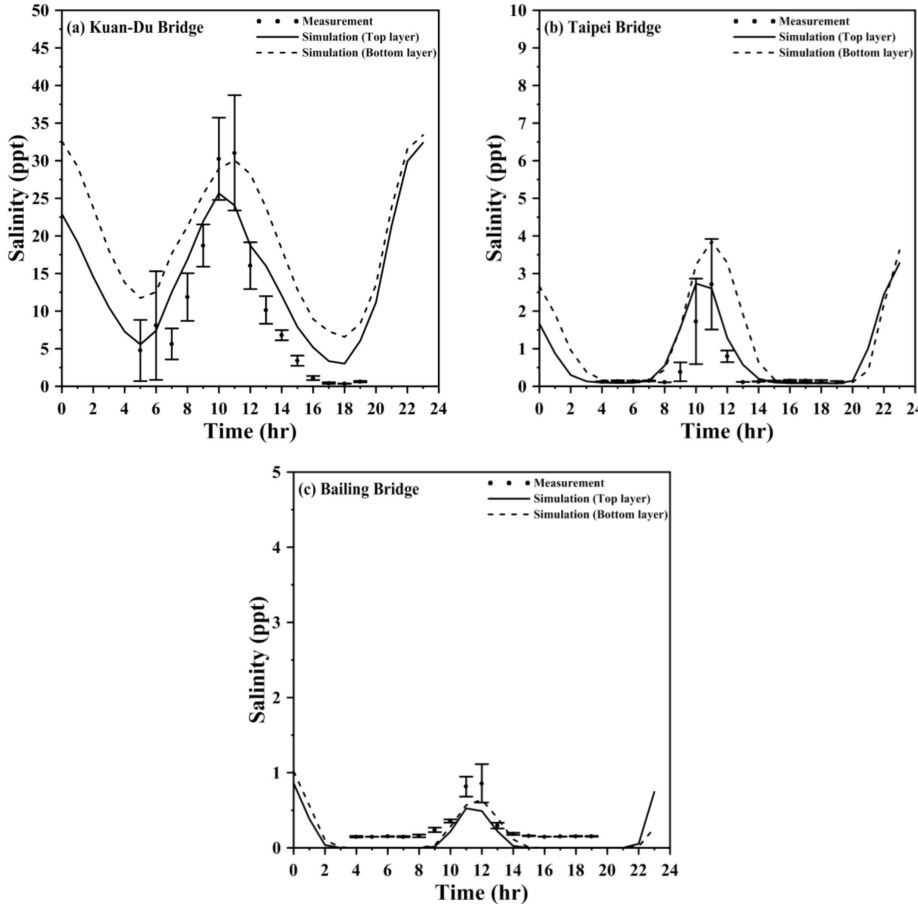

**Figure 10.** Comparison of the simulated salinity with time-series data on 24 June 2017 at the (**a**) Kuan-Du Bridge, (**b**) Taipei Bridge, and (**c**) Bailing Bridge (model validation).

## 5. Results

The validated model was utilized to carry out several runs to explore hydrodynamic characteristics, salinity distribution, and residence time under the present and predevelopment conditions. The earliest available data for topography and bathymetry were in 1981, which represents the predevelopment condition, and the topographic and bathymetric data were collected in 2015, which is regarded as the present condition (see Figure 4).

### 5.1. Water Levels and Residual Currents

Model simulations were conducted using five constituent tides at the sea boundary to yield the water surface elevation in the time series to drive the model simulations. The freshwater discharges were denoted using low flow at the upstream boundaries of three major tributaries. A $Q_{75}$ flow equal to or exceeding 75% of time is recognized as the low-flow condition. The $Q_{75}$ flow is a common low-flow design used in Taiwan's rivers. The $Q_{75}$ freshwater discharges at the three tributaries were 3.74 m³/s, 11.6 m³/s, and 3.5 m³/s for the Dahan Stream, the Hsintien Stream, and the Keelung River, respectively. A constant salinity of 35 ppt at the open sea boundary and 0 ppt at the upstream boundaries of the three major tributaries was set up for the model simulations.

Figure 11 shows the calculated water surface elevations at five stations under the predevelopment conditions and present conditions. The water surface elevation under the present condition was higher than under the predevelopment condition at high tide but was lower at low tide, which resulted in a

tidal range under the present condition that was higher than the range under the predevelopment condition. The time lag at both high tide and low tide under the present condition was approximately 0.5 h, 1 h, 2 h, 2 h, and 1 h, shorter than the time lag under the predevelopment condition at the Kuan-Du Bridge, Taipei Bridge, Hsin-Hai Bridge, Zhong-Zheng Bridge, and Bailing Bridge, respectively. The time lag propagated from downstream to the upstream reaches, resulting in a 2-h lag at the Hsin-Hai Bridge and Zhong-Zheng Bridge. Through the model predictions, we found that the changes in bathymetry and topography from 1981 to the present had a strong influence on tidal wave propagation.

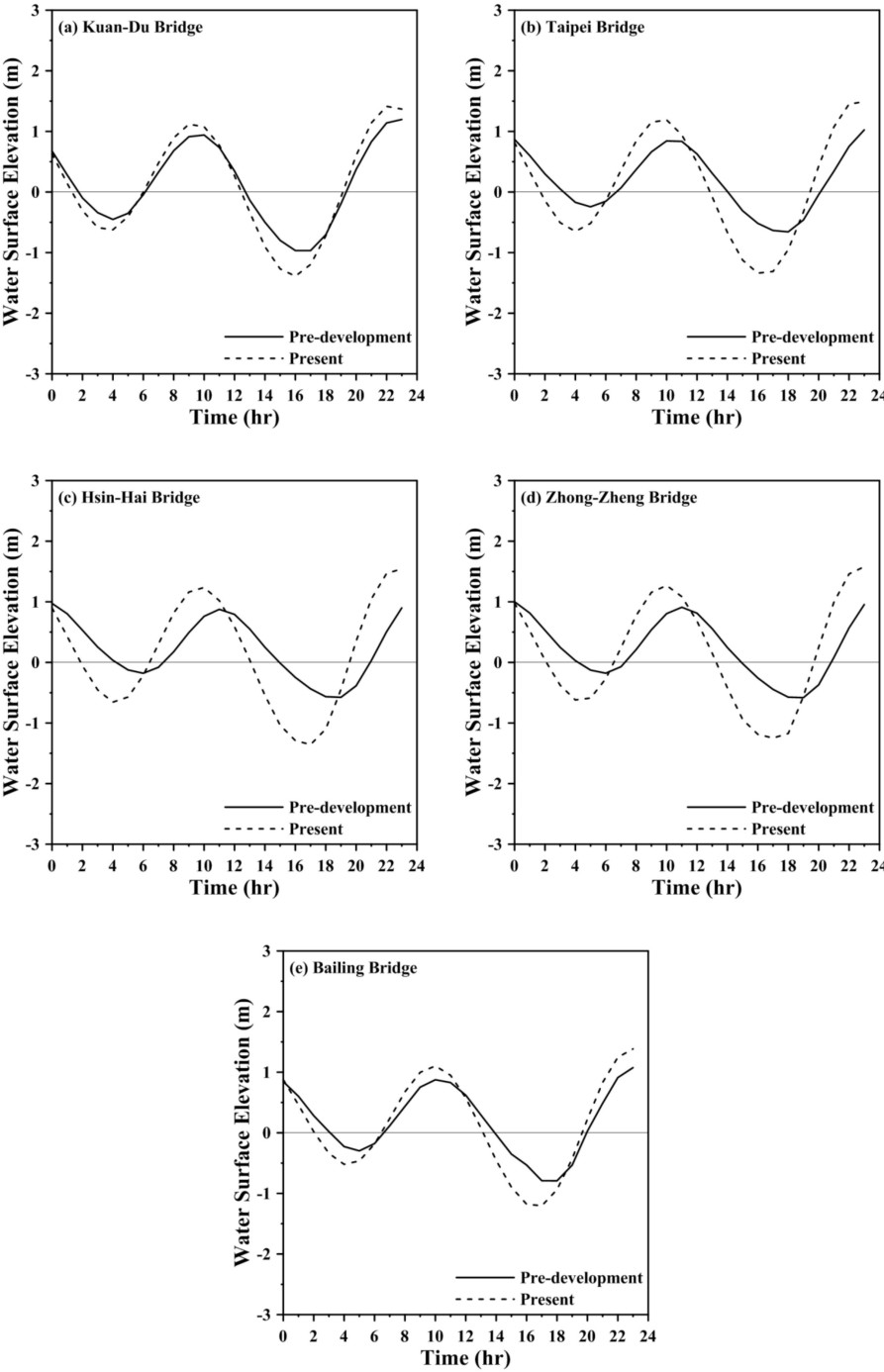

**Figure 11.** Comparison of the water surface elevation under the predevelopment and present conditions for Q₇₅ low flow at the (**a**) Kuan-Du Bridge, (**b**) Taipei Bridge, (**c**) Hsin-Hai Bridge, (**d**) Zhong-Zheng Bridge, and (**e**) Bailing Bridge.

The current in tidal estuaries can be divided into two types: Tidal currents and residual currents. Traditionally, the scale is 0.1 m/s for the residual current (or the subtidal current), compared to 1 m/s for the tidal current [44]. The averaged velocities over two spring-neap cycles were calculated from the model simulation results to yield the residual currents.

Figures 12 and 13 show the computed residual current under the predevelopment and present conditions, respectively, in the estuarine system. These two figures indicate that the deeper channel enhanced the residual circulation that pushed salinity further upriver. Under the $Q_{75}$ low-flow condition, apparent residual circulation occurred under both the predevelopment and present conditions in the estuarine system. However, the residual circulation under the predevelopment condition was stronger than under the present condition.

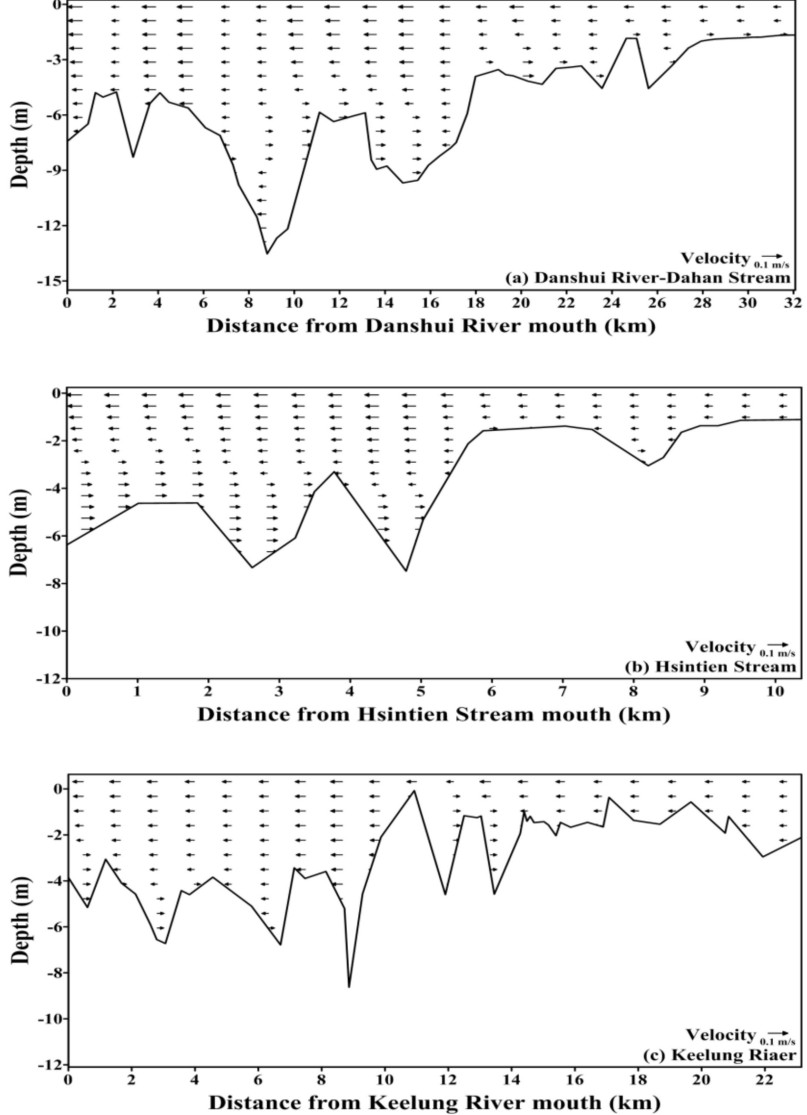

**Figure 12.** Computed residual current under the predevelopment condition for $Q_{75}$ low flow in the (**a**) Danshui River–Dahan Stream, (**b**) Hsintien Stream, and (**c**) Keelung River.

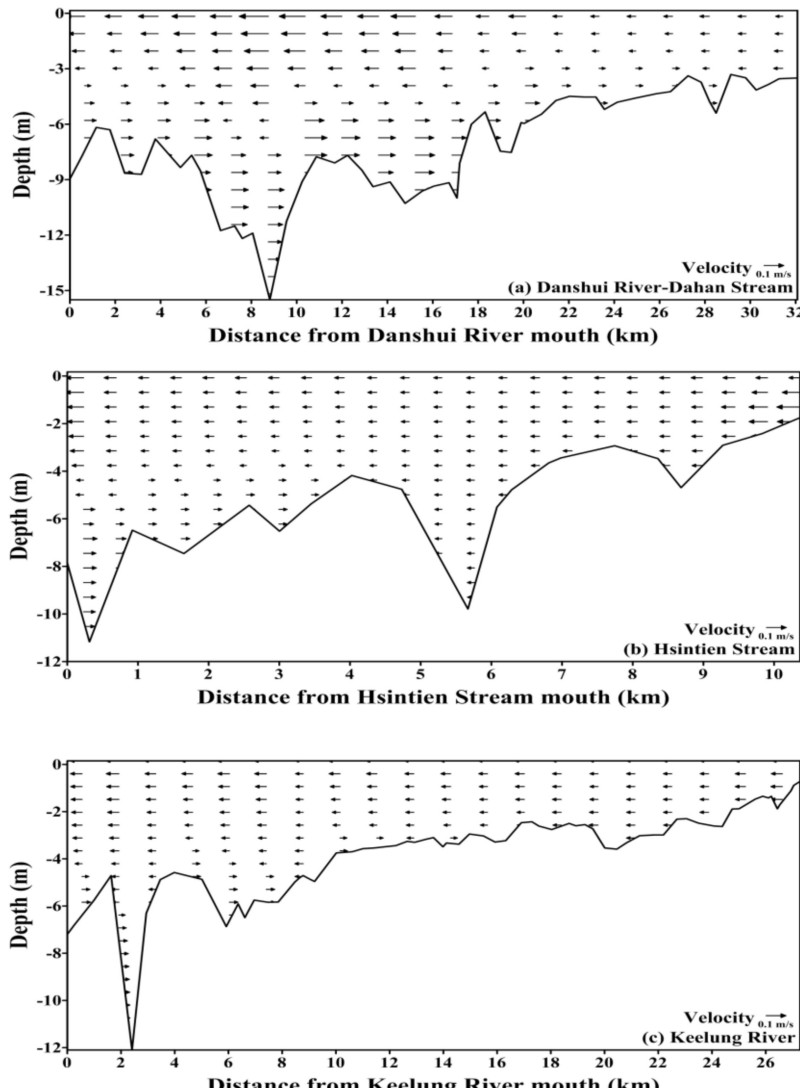

**Figure 13.** Computed residual current under the present condition for $Q_{75}$ low flow in the (**a**) Danshui River–Dahan Stream, (**b**) Hsintien Stream, and (**c**) Keelung River.

*5.2. Saltwater Intrusion*

Spatial variations in salinity concentration present basic knowledge for studying estuarine dynamics. Salinity plays an important role in estuarine research and serves as a water quality variable utilized for ecological research and management. Changing the salinity distribution would significantly influence the ecological balance in estuarine systems [45].

The model was run under various flow conditions at upstream boundaries to investigate the salinity distribution and the limit of salt intrusion. Except for the $Q_{75}$ flow condition, the other scenarios of freshwater discharges imposed at upstream boundaries are listed in Table 4. The long-term historically daily freshwater discharges in three main tributaries were analyzed to yield the flow equal to or exceeding percentage of time. Even though the flows presented in Table 4 could not represent the realistic hydrograph conditions, they were obtained from the statistical analysis results to cover different scenarios.

**Table 4.** Freshwater discharges imposed at upstream boundaries.

| Flow Condition | Dahan Stream (m³/s) | Hsintien Stream (m³/s) | Keelung River (m³/s) |
|---|---|---|---|
| $Q_{10}$ | 64.78 | 131.36 | 67.00 |
| $Q_{20}$ | 31.79 | 79.40 | 35.66 |
| $Q_{30}$ | 19.33 | 53.80 | 21.50 |
| $Q_{40}$ | 12.55 | 38.37 | 14.00 |
| $Q_{50}$ | 8.52 | 27.10 | 9.53 |
| $Q_{60}$ | 5.98 | 20.08 | 6.46 |
| $Q_{70}$ | 4.32 | 14.22 | 4.38 |
| $Q_{80}$ | 3.19 | 9.01 | 2.72 |
| $Q_{90}$ | 2.19 | 3.76 | 1.33 |

Figures 14 and 15 display the predicted salinities in the spatial distribution for $Q_{75}$ low flow under the predevelopment and present conditions, respectively, in the estuarine system. Salinity is averaged over 58 tidal cycles. The simulated results indicated that salinity exhibited stratification in the three major tributaries. Compared to the predicted salinities for the low-flow condition shown in Figures 14 and 15, the limit of salt intrusion was further upstream under the predevelopment condition compared to the limit of salt intrusion under the present condition in the Danshui River–Dahan Stream and Hsintien Stream, but the distance of salt intrusion was quite similar in the Keelung River. However, the salinity in the downstream reaches of the Keelung River was high under the predevelopment period compared to under present condition. Therefore, the decreasing salinity under the present condition as a result of bathymetric and topographic changes might not be the reason to alter the aquatic ecosystem in the lower reaches of tidal estuaries.

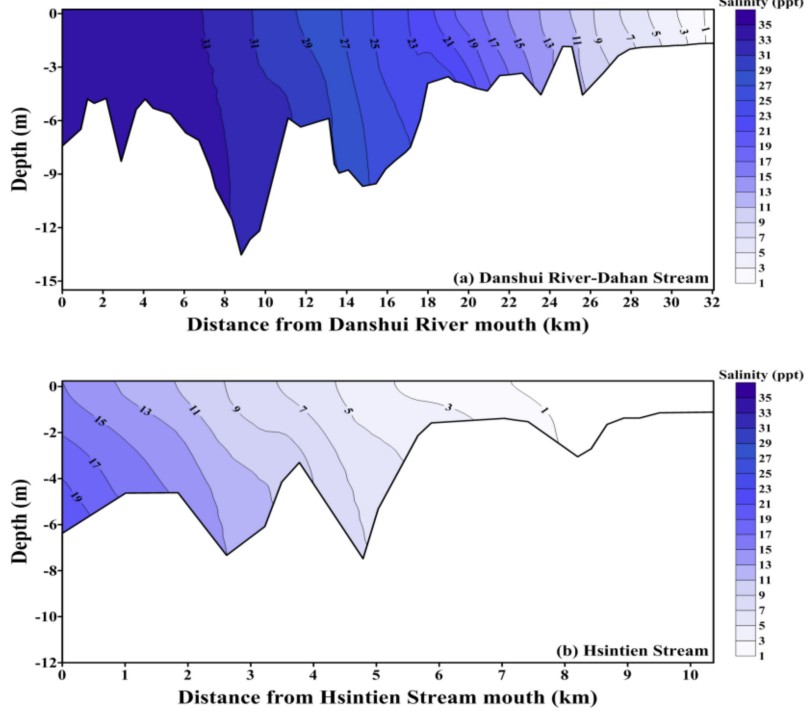

**Figure 14.** *Cont.*

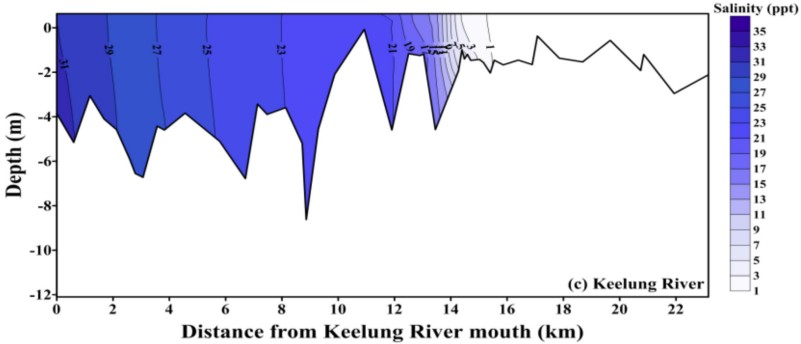

**Figure 14.** Computed tidally averaged salinity distribution under the predevelopment condition for Q$_{75}$ low flow in the (**a**) Danshui River–Dahan Stream, (**b**) Hsintien Stream, and (**c**) Keelung River.

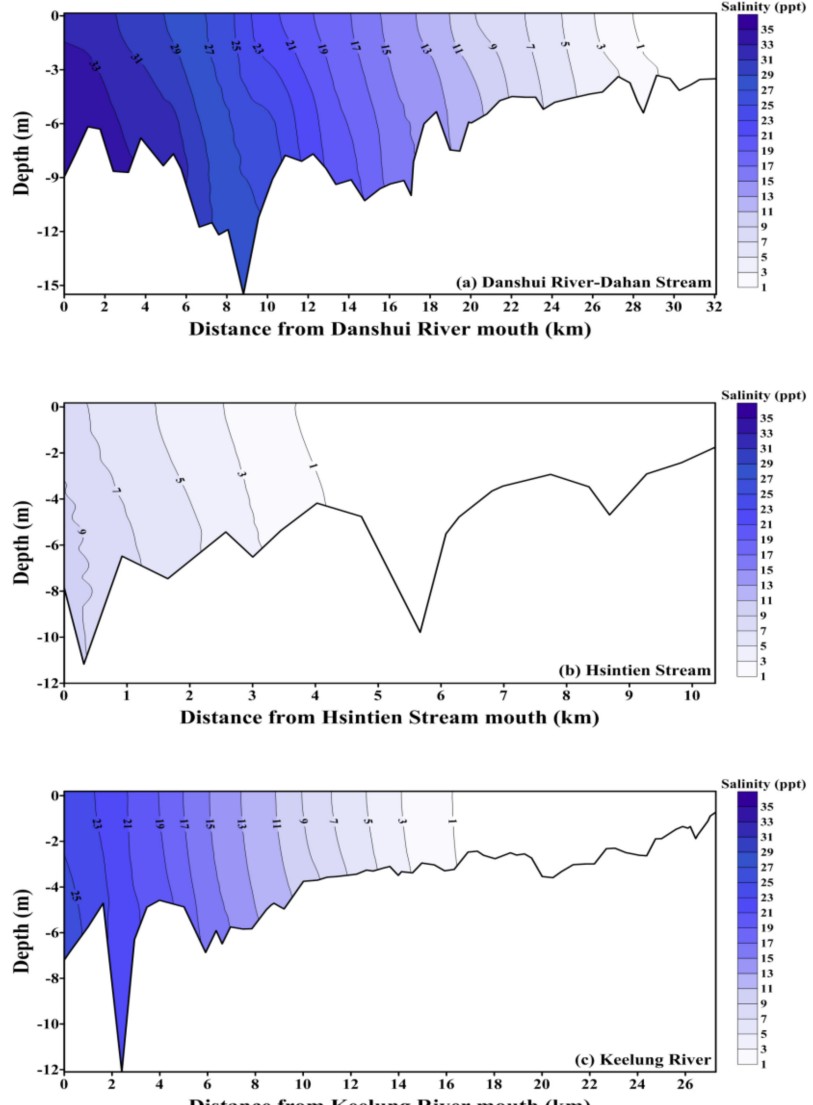

**Figure 15.** Computed tidally averaged salinity distribution under the present condition for Q$_{75}$ low flow in the (**a**) Danshui River–Dahan Stream, (**b**) Hsintien Stream, and (**c**) Keelung River.

The limit of salt intrusion for different inputs of freshwater discharge from upstream boundaries is shown in Figure 16. The regression fit was developed to predict the maximum distance of salt intrusion as a function of freshwater discharge flowing into each tributary. The regressions gained the maximum distance of salinity intrusion ($D_{SI}$) for a known freshwater discharge ($Q$) at Danshui River–Dahan Stream:

$$D_{SI} = 34.98e^{-0.014Q} \; r^2 = 0.97 \text{ for predevelopment condition} \tag{3}$$

$$D_{SI} = 27.90e^{-0.014Q} \; r^2 = 0.88 \text{ for present condition} \tag{4}$$

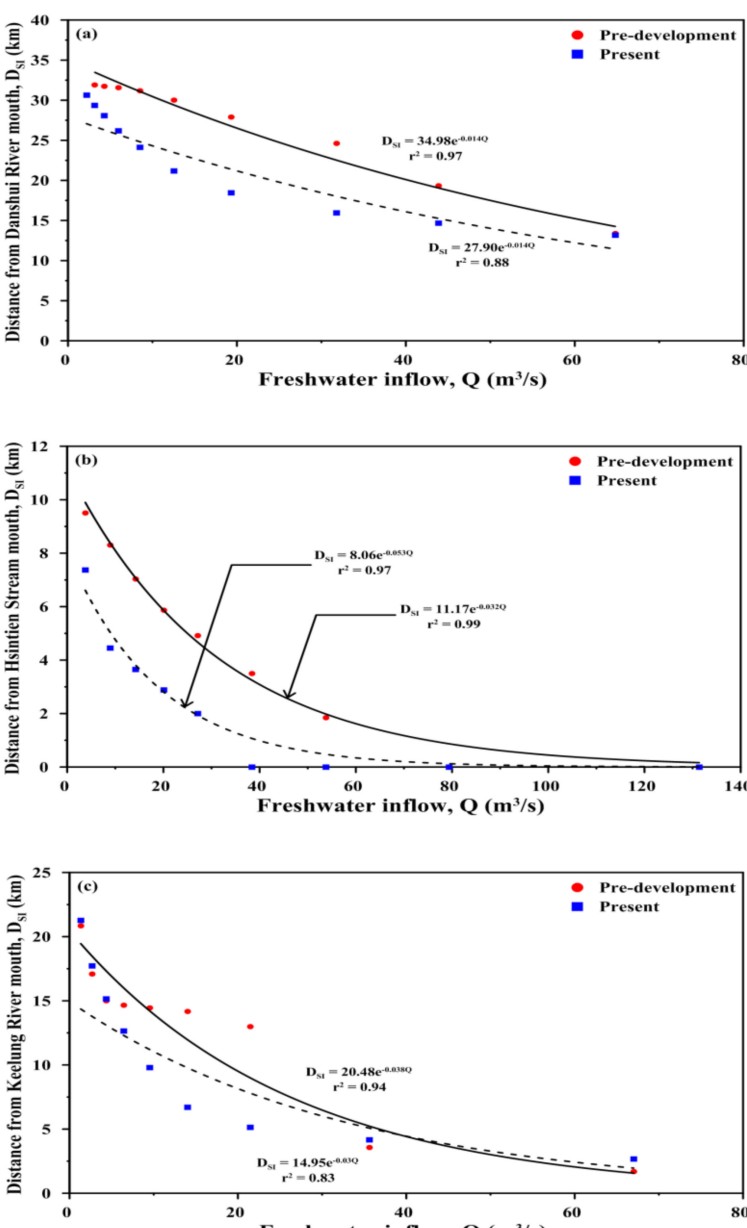

**Figure 16.** Relationship between the limit of salt intrusion and freshwater discharge under the predevelopment and present conditions in the (**a**) Danshui River–Dahan Stream, (**b**) Hsintien Stream, and (**c**) Keelung River.

Hsintien Stream:

$$D_{SI} = 11.17e^{-0.032Q} \ r^2 = 0.99 \text{ for predevelopment condition} \tag{5}$$

$$D_{SI} = 8.06e^{-0.053Q} \ r^2 = 0.97 \text{ for present condition} \tag{6}$$

Keelung River:

$$D_{SI} = 20.84e^{-0.038Q} r^2 = 0.94 \text{ for predevelopment condition} \tag{7}$$

$$D_{SI} = 14.95e^{-0.03Q} r^2 = 0.83 \text{ for present condition} \tag{8}$$

*5.3. Residence Time*

The residence time can act as an index to represent the transport time scale of physical, chemical, and biochemical processes. The concept of residence time has been widely used to study water exchange in estuarine waters. In this study, two approaches, Sanford et al. [46] in Equation (9) and Luketina et al. [47] in Equation (10), were used to calculate the residence time to represent the time requested for the total mass of the conservative tracer within the estuary to be decreased as a factor of the e-folding value ($e^{-1}$ value). The equations to estimate the residence time ($T_r$) can be expressed as follows:

$$T_r = \frac{(V + \frac{P}{2})T}{(1-b)P + QT} \tag{9}$$

$$T_r = \frac{(V + P)T}{(1-b)P + \frac{QT}{2}} \tag{10}$$

where $b$ denotes the return flow, $P$ expresses the tidal prism, and $T$ represents the tidal period.

To determine the return flow ($b$), the case without river discharge from upstream boundaries was run with the model. The return flow was calculated to be 0.83 and 0.78 under the predevelopment and present conditions, respectively, for Equation (9), while the return flow was calculated to be 0.74 and 0.67 under the predevelopment and present conditions, respectively, for Equation (10). Then, Equations (9) and 10) can be applied to calculate the residence times for various freshwater discharges. Figures 17 and 18 show the residence time vs. freshwater discharge under the predevelopment and present conditions using Equations (9) and 10), respectively. The regression relationship between residence time ($T_r$) and freshwater discharge ($Q$) has been documented with an excellent coefficient of determination ($r^2$) to form the equation ($T_r = ae^{-bQ}$, where a and b are constant values) [48,49]. The regression yielded with Equation (9),

$$T_r = 73.78e^{-0.002Q} \ r^2 = 0.99 \text{ for predevelopment condition} \tag{11}$$

$$T_r = 54.54e^{-0.002Q} \ r^2 = 0.99 \text{ for present condition} \tag{12}$$

and with Equation (10),

$$T_r = 73.35e^{-0.0009Q} \ r^2 = 0.99 \text{ for predevelopment condition} \tag{13}$$

$$T_r = 54.69e^{-0.0009Q} \ r^2 = 0.99 \text{ for present condition} \tag{14}$$

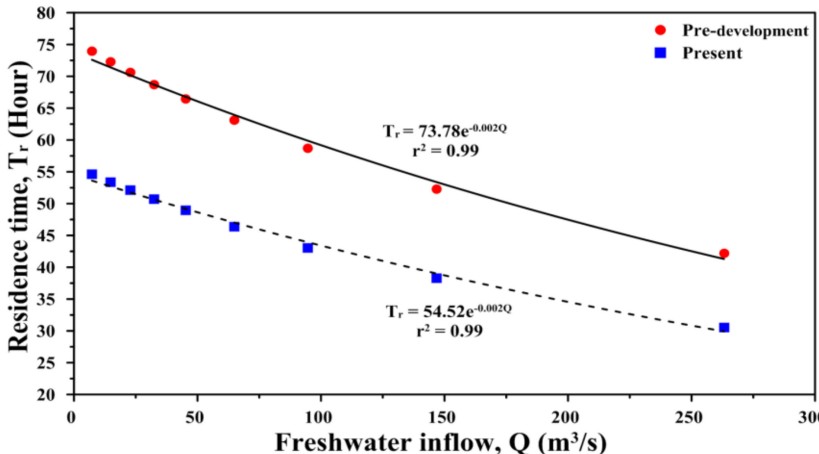

**Figure 17.** Relationship between the residence time and freshwater discharge using Sanford et al.'s [46] approach under the predevelopment and present conditions.

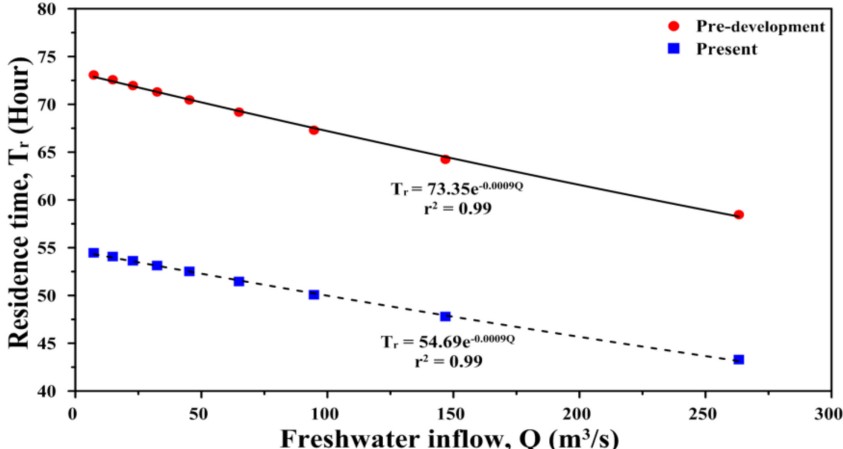

**Figure 18.** Relationship between the residence time and freshwater discharge using Luketina et al.'s [47] approach under the predevelopment and present conditions.

The modeling results indicated that when the freshwater discharge input increased, the residence time concurrently decreased. The calculated residence time using Equations (9) and 10) was the same order of magnitude, approximately 1–3 days, depending on freshwater discharge. Importantly, the residence time under the predevelopment condition was longer than under the present condition.

## 6. Discussion

To explore the influence of geomorphic-driven changes on tidal prism, different scenarios of freshwater discharges imposed at upstream boundaries were used to calculated tidal prism. The equation of Lakhan [50] was adopted to yield the tidal prism. The equation is expressed as, $P = H \cdot A$, where $P$ expresses the tidal prism, $H$ denotes the mean tidal range, and $A$ represents the surface area. The results revealed that the tidal prism for $Q_{10}$ high flow condition was $8.91 \times 10^7$ m$^3$ under the predevelopment condition, decreasing to $6.45 \times 10^7$ m$^3$ under the present condition. The tidal prism for $Q_{75}$ low flow condition under the predevelopment condition was $8.8 \times 10^7$ m$^3$, decreasing to $6.40 \times 10^7$ m$^3$ under present condition. We found that the tidal prism under the present condition was less than that under predevelopment condition as a result of geomorphic-driven changes. In the channel, the water volume below the mean sea level under predevelopment condition was also larger than that under present condition.

According the simulated results of water surface elevation, a time lag of 0.5–2 h was found at different gauge stations. Similar results regarding the effect of bathymetric changes on time lag were also documented in Madalena et al. [23], Picado et al. [13], and Passeri et al. [24].

Changes in residual circulation were known to crucially affect saltwater intrusion and residence time [28,51,52]. Meyers et al. [53] documented that deepening the channel increased the magnitude of the residual circulation in a coastal plain estuary. Local bathymetry and morphology were important to alter the residual circulation. Our study found that some regions within the estuary were deepened under the predevelopment condition compared to the present condition, resulting in an increase in the residual circulation under the predevelopment condition, thus altering the salt intrusion and residence time in estuarine system.

Several studies have been carried out on this topic regarding the limits of salt intrusion and freshwater discharge through observational data analysis or numerical prediction [54–57]. In this study, the high coefficient of determination presented in Equations (3)–(8) revealed that freshwater discharge from upstream boundaries played an important role in determining the limit of salt intrusion.

The calculated residence time under present condition was shorter than that under predevelopment condition. A short residence time would be beneficial to pollutant removal in estuaries. Based on the model prediction, the current condition was conducive to the discharge of pollutants from the estuary offshore. Wang et al. [58] calculated residence times ranging from one to two days under different scenarios of freshwater discharge in the Danshui River estuary. For the present condition, the residence time computed by this study is close to the literature reported by Wang et al. [58].

To analyze the impacts of geomorphic-driven changes on hydrodynamics, saltwater intrusion, and residence time in tidal estuarine system, we adopted the same value of bottom roughness height under predevelopment and present conditions to assume the same bottom friction because no observed data were used to calibrate and validate the model under predevelopment condition. This assumption would be produced some biases in model simulation under predevelopment condition. However we believed that the simulated results and patterns would not have much change. The wind-wave action is not included in current study. Future recommendation is to consider the wind-wave effect, especially when the sediment transport is included in the model [59].

## 7. Conclusions

A 3D SELFE-based model was exploited for a case study in the Danshui River estuarine system to investigate the influence of bathymetric and topographic changes on hydrodynamics, salt distribution, and residence time. The numerical model was calibrated and validated using the observational water level, tidal current, and salinity data from 2015, 2016, and 2017. Overall, the simulated results quantitatively reproduced the measured data

The well-calibrated and validated model was further utilized to investigate the hydrodynamics, residual current, salinity distribution, limit of salt intrusion, and residence time under the predevelopment and present conditions. The simulated results indicated that the time lag at both high tide and low tide under the present condition was approximately 0.5–2 h shorter than under the predevelopment condition at the five gauge stations. Local bathymetry and morphology were critical for changing the residual circulation. Some regions within the estuary were deepened under the predevelopment condition compared to under the present condition, resulting in an increase in the residual circulation under the predevelopment condition. The limit of salt intrusion was pushed further upstream under the predevelopment condition compared to under the present condition in the Danshui River–Dahan Stream and Hsintien Stream, but the distance of salt intrusion was similar in the Keelung River. However, the salinity distribution in the downstream reaches of the Keelung River was higher under the predevelopment condition than under the present condition. Furthermore, a regression fit was developed to predict the maximum distance of salt intrusion as a function of freshwater discharge flowing into the estuarine system.

Two approaches—Sanford et al. [45] and Luketina et al. [46]—were adopted to calculate the residence time under the predevelopment and present conditions. An analysis of the regression correlation between residence time and freshwater discharge was developed to understand the hydrological and physical processes in estuarine systems. We found that the residence time under the predevelopment condition was longer than under the present condition. This result suggests that a shorter residence time is helpful for discharging pollutants into the coastal sea under the present condition.

This study provides fundamental knowledge regarding how bathymetric and topographic changes in tidal estuaries affect hydrodynamics, salt intrusion, and residence time. Further research will incorporate a sediment transport model into a hydrodynamic model to probe the influence of geomorphic-driven changes on suspended sediment transport. In the future work, the sensitivity tests will be executed to understand the sensitivity of processes to geomorphic changes.

**Author Contributions:** Conceptualization, W.-C.L.; Methodology, W.-C.L. and H.-M.L.; Software, M.-H.K. and H.-M.L.; Validation, W.-C.L., M.-H.K. and H.-M.L.; Investigation, W.-C.L.; Resources, W.-C.L.; Writing—Original draft preparation, W.-C.L.; Writing—Review and editing, W.-C.L.; Supervision, W.-C.L. and H.-M.L.; Funding acquisition, W.-C.L. All authors have read and agreed to the published version of the manuscript.

**Funding:** This present study was partially funded by the Taiwan Ministry of Science and Technology (MOST 105-2625-M-239-MY2).

**Acknowledgments:** The authors express sincere acknowledgments to 10th River Management Office, TWRA for offering the valuable prototype data. The authors would like to express their appreciation to two anonymous reviewers who provided useful comments to substantially improve this manuscript.

**Conflicts of Interest:** The authors declare no conflict of interest.

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
