# Peer review of "Response of Salt Transport and Residence Time to Geomorphologic Changes in an Estuarine System"

_water, doi:10.3390/w12041091_

Round 1

Reviewer 1 Report

In the “Introduction” part, the authors should describe the scientific questions and significance associated with their investigation and comparison of predevelopment and present estuarine conditions. The authors need to tell the audience why they do comparison of these two conditions and why this comparison is important. The authors also need to add the literature review related to their scientific questions.

On Line 197-198, can the authors explain the water level, velocity and salinity values that they chose for initial conditions? For a typical hydrodynamic model setup, the rest condition (i.e. water level 0 m and velocity 0 m/s) and the climatological salinity field (e.g. World Ocean Atlas) are used as initial conditions. Have the authors tested the typical hydrodynamic model setup and compared with their current initial conditions? Have the authors done the ramp-up run to allow the model reaching an equilibrium state before they compared the model results with the observations?  

On Line 235 Figure 5, there are some biases (i.e. differences in mean water level) between modeled and observed water levels especially for all stations except (a). Can the authors add some explanation and discussion here?

On Line 302 Figure 9, for the stations (d) and (e), the modeled velocities underestimate the observed velocities. Can the authors add some explanation and discussion here?

Author Response

Responses to Reviewers’ Comments

Reviewer 1:

In the “Introduction” part, the authors should describe the scientific questions and significance associated with their investigation and comparison of predevelopment and present estuarine conditions. The authors need to tell the audience why they do comparison of these two conditions and why this comparison is important. The authors also need to add the literature review related to their scientific questions.

Response: We deeply appreciate the valuable comments by Reviewer. Text and citations have been included to respond this comment. (Page 2, lines 73-77)

Increasing the distance of salt intrusion led to changes in the ecosystem and increasing the residence time of the water body caused more pollution to stay in the estuary for too long [15,16,23,28,29]. Therefore there is urgent need to better understanding the effects on geomorphic-driven changes on salt water intrusion and residence time in estuarine system.”

On Line 197-198, can the authors explain the water level, velocity and salinity values that they chose for initial conditions? For a typical hydrodynamic model setup, the rest condition (i.e. water level 0 m and velocity 0 m/s) and the climatological salinity field (e.g. World Ocean Atlas) are used as initial conditions. Have the authors tested the typical hydrodynamic model setup and compared with their current initial conditions? Have the authors done the ramp-up run to allow the model reaching an equilibrium state before they compared the model results with the observations?  

Response: Text has been added to response this comment. (Page 7, lines 237-242)

“The initial conditions adopted were mean values of 1.5 m, 0.6 m/s, and 25 ppt for the water level, velocity, and salinity, respectively, utilized to warm up the model. We also tested the rest condition (i.e. water level 0 m and velocity 0 m/s) and the climatological salinity field used as initial conditions. However, the 30-day warm up time was needed to reach an equilibrium state before the model results were compared with the observations.”

On Line 235 Figure 5, there are some biases (i.e. differences in mean water level) between modeled and observed water levels especially for all stations except (a). Can the authors add some explanation and discussion here?

Response: Text has been added to explain the some biases. (Page 9, lines 277-279)

The underestimated water surface elevations during high tide would be the reason that the daily freshwater discharges were used as upstream boundaries in three major tributaries.”

On Line 302 Figure 9, for the stations (d) and (e), the modeled velocities underestimate the observed velocities. Can the authors add some explanation and discussion here?

Response: Text has been added to explain the possible reason. (Page 14, lines 340-344)

It is the reason that the Zhong-Zheng Bridge is close to the upstream boundary and is not subject to tidal effect. The interpolation of daily freshwater discharge at upstream boundary can not reflect the hourly flow. The underestimated measured results during ebb tide at the Bailing Bridge would be the reason that the roughness height used at this station was not suitable.”

Reviewer 2 Report

This was a well presented model case study of an estuarine system that has been impacted by geomorphic changes.
The model methodology seemed rigorous with appropriately developed models and good validation.
  My main comment is that the authors should find quantitative ways for the reader to be able to infer general outcomes from this case study.
As a first step, the level of geomorphic change should be quantified in a general sense.
Also an appreciation of the natural geomorphic variability within the system should be discussed.
Then possibly sensitivity tests should be done to understand the sensitivity of the processes to the geomorphic changes.
This would involve other configurations of the model.
  Abstract:
The geomporphic-driven changes in...
Faitfully mimicked = corresponded well to
Need to say what predevelopment conditoions are
Time lag of what?
  Introduction:
The introduction is short. It basically states a lack of knowlege on the impacts of coastal modifications, but does not really go into why this is an issue.
It describes a few example studies but does not really summarise the fundamental science that has been established in this field and what hasn't - hence there is not a clear aim for the paper.
Only an intent to model a case study system.
I think the authors should show how this study will advance our general understanding of geomorphic-driven hydrodynamics in estuaries.
L32:  ...that are subject to...
L50: elaborate on 'further ecological management purposes'. Need to explain why salt intrusion, circulation and residence are important (i.e much more than just ecological managemnet).
  Site description:
First paragraph: Can you describe the temporal variability on river flow, and the data resolution and location available.
Also what is the surface area and volume of the estuary?
2nd paragraph: What is the tidal prism? What is the significance of the standing wave system and does the phase lag change further up-estuary?
Is there tidal asymmetry (e.g. flood dominance causing sediment influx)?
Line 101: Make sure 'material' refers to neutrally buoyant material and not denser sediment.
What is the sediment makeup of the estuary?
Is there an influence of waves at the mouth?
  Methods:
3.2: Ned to describe the time period, spatial and temporal resolution of the bathymetry data.
3.3: where do the tidal forcing come from? And resolution?
Calibration and validation:
Need to state the location and spatial and temporal resolution of the data used for validation.
  Discussion:
This is really a 'Results' section so change the name.
I think there needs to be a more quantified description of the development changes to the system.
For example, overall, by how much has the estuary volume changed? And the widths and depths?
This information will help readers interpret your results more generally.
L364: You ran the model under several flow conditions, but what would happen under more realistic hydrograph conditions, such as a peak flow event rather than constant flow?
  A separate discussions section should be presented that generalises the results and puts them into a general or global context.
And assumptions and future recommendations in this field.

Author Response

Responses to Reviewers’ Comments

Reviewer 2:

This was a well presented model case study of an estuarine system that has been impacted by geomorphic changes.
The model methodology seemed rigorous with appropriately developed models and good validation. My main comment is that the authors should find quantitative ways for the reader to be able to infer general outcomes from this case study.
As a first step, the level of geomorphic change should be quantified in a general sense.
Also an appreciation of the natural geomorphic variability within the system should be discussed. Then possibly sensitivity tests should be done to understand the sensitivity of the processes to the geomorphic changes. This would involve other configurations of the model.  

Response: We deeply appreciate the valuable comments by Reviewer. As the Reviewer mentioned that this is first step to take into account the impacts of geomorphic-driven change on hydrodynamics, saltwater intrusion, and residence time in tidal estuarine system, further work would involve different geomorphologies for different periods to understand the sensitivity of the processes to geomorphic changes. Text has been added in the revised manuscript. (Page 26, lines 548-549)

“In the future work, the sensitivity tests will be executed to understand the sensitivity of processes to geomorphic changes.”

Abstract:
The geomorphic-driven changes in...

Response: It has been revised. (Page 1, line 10)

The geomorphic-driven changes in estuarine hydrodynamics and salt transport remain unclear.”

Faithfully mimicked = corresponded well to

Response: It has been revised. (Page 1, line 14)

“The performance of the SELFE model corresponded well to the measured data.”

Need to say what predevelopment conditions are

Response: The predevelopment has been defined in the revised manuscript. (Page 1, lines 14-16)

“Furthermore, the validated model was utilized to analyze the hydrodynamics, residual current, limit of salt intrusion, and residence time under the predevelopment (1981) and present (2015) conditions.”

Time lag of what?  

Response: It has been revised. (Page 1, lines 16-18)

“The predicted results revealed that the time lag of water surface elevation at both high tide and low tide under the present condition was approximately 0.5-2 hours shorter under the predevelopment condition.”

Introduction:
The introduction is short. It basically states a lack of knowlege on the impacts of coastal modifications, but does not really go into why this is an issue.

Response: Text has been added to respond this comment.

It is worth noting that river modifications, such as channel deepening and widening, dredging, jetty construction, tidal flat and marsh filling, and river realignment [10-17], have been carried out over the past several decades; therefore, understanding the long-term responses to salt transport, estuarine flows, and residence time in tidal estuaries as a result of anthropogenic activities is of utmost importance for further ecological management purposes, because the salt intrusion, estuarine circulation, and residence time crucially affect the water quality and sediment transport, fecal coliform, and heavy metal transport [18-22].” (Page 2, lines 47-53)

A number of publications have documented that anthropogenic modifications alter physical processes and possibly change estuarine ecosystems. Increasing the distance of salt intrusion may led to changes in the ecosystem and increasing the residence time of the water body caused more pollution to stay in the estuary for too long [15,16,23,28,29]. Therefore there is urgent need to better understanding the effects on geomorphic-driven changes on salt water intrusion and residence time in estuarine system.” (Page 2, lines 72-77)

It describes a few example studies but does not really summarise the fundamental science that has been established in this field and what hasn't - hence there is not a clear aim for the paper. Only an intent to model a case study system. I think the authors should show how this study will advance our general understanding of geomorphic-driven hydrodynamics in estuaries.

Response: Text has been added in the revised manuscript. (Page 2, lines 72-83)

A number of publications have documented that anthropogenic modifications alter physical processes and possibly change estuarine ecosystems. Increasing the distance of salt intrusion led to changes in the ecosystem and increasing the residence time of the water body caused more pollution to stay in the estuary for too long [15,16,23,28,29]. Therefore there is urgent need to better understanding the effects on geomorphic-driven changes on salt water intrusion and residence time in estuarine system. The primary objective of the present study is to analyze the hydrodynamics, salt intrusion, and residence time in the Danshui River estuarine system under the predevelopment (1981) and present conditions.”

L32:  ...that are subject to...

Response: It has been revised. (Page 1, lines 33-34)

L50: elaborate on 'further ecological management purposes'. Need to explain why salt intrusion, circulation and residence are important (i.e. much more than just ecological management).  

Response: Text has been added to respond this comment. (Page 2, lines 49-53)

therefore, understanding the long-term responses to salt transport, estuarine flows, and residence time in tidal estuaries as a result of anthropogenic activities is of utmost importance for further ecological management purposes, because the salt intrusion, estuarine circulation, and residence time crucially affect the water quality and sediment transport, fecal coliform, and heavy metal transport [18-22].”

Site description:
First paragraph: Can you describe the temporal variability on river flow, and the data resolution and location available.

Response: Text has been added to address the comment in the revised manuscript. (Pages 2-3, lines 93-96)

The daily freshwater discharges at the Cheng-Ling Bridge, Hsiu-Lang Bridge, and Wu-Tu stations were gathered for flow analysis. The mean river discharges are 39.0 m3/s, 69.7 m3/s, and 25.0 m3/s, respectively, in the Dahan Stream, Hsintien Stream, and Keelung River, while the low river discharges are 2.2 m3/s, 3.8 m3/s, and 1.3 m3/s.”

Also what is the surface area and volume of the estuary?

Response: Text has been added to describe the volume and surface area. (Page 3, lines 96-97)

“The average annual flow rate and surface area are 6.6x109 m3 and 1.74x109 m2 approximately, respectively [31].”

2nd paragraph: What is the tidal prism? What is the significance of the standing wave system and does the phase lag change further up-estuary? Is there tidal asymmetry (e.g. flood dominance causing sediment influx)?

Response: Text has been added in the revised manuscript.

During the normal flow condition, the water volume between the highest tide level and the lowest tide level called the tidal prism is about 6.5x107 m3 [32].” (Page 3, lines 103-104)

“The standing wave affects the material transport in tidal estuaries. No phase difference was found at the upstream reaches where were beyond the tidal imit. Chen and Liu [32] demonstrated that a marked tidal asymmetry occurred in the Danshui River estuary.” (Page 3, lines 107-110)

Line 101: Make sure 'material' refers to neutrally buoyant material and not denser sediment. What is the sediment makeup of the estuary?

Response: Text has been added to respond this comment. (Page 3, lines 117-122)

Residence time represents a vital index for assessing the material transport (i.e. neutrally buoyant material) of estuaries. Generally, this index is adopted to evaluate the removal rate of estuarine pollutants transported by river discharge. During the normal weather condition, the influence of waves at the mouth is not important. Therefore the wave action is not included in the model simulations. The sediment transport of the Danshui River comes from the upstream watershed. The annual sediment yield in the Danshui River is about 1.145x106 t [34].”

Is there an influence of waves at the mouth?  

Response: Text has been added to address the comment. (Page 3, lines 119-121)

“During the normal weather condition, the influence of waves at the mouth is not important. Therefore the wave action is not included in the model simulations.”

Methods:
3.2: Need to describe the time period, spatial and temporal resolution of the bathymetry data.

Response: Text has been added to respond the comment. (Page 6, lines 192-197)

The modeling domain was extended from the estuarine system to the continental shelf (Figure 4). To establish the meshes for running the model, bathymetric and topographic data measured in 1981 and 2015 from the estuarine system and continental shelf were gathered from government agencies in Taiwan. A field survey was conducted by the TWRA to measure the cross-sectional profiles every 300-500m along the river. The resolution of 10 km for bathymetric and topographic data in the coastal sea was measured by the National Center for Ocean Research.”

3.3: where do the tidal forcing come from? And resolution?

Response: Text has been added to response this comment. (Page 7, lines 233-236)

“The time-series water level was generated by five tidal components (i.e., M2, S2, N2, K1, and O1) specified at the open boundaries to drive the model run. Five tidal components with resolution of 5 km in coastal sea were obtained from the National Center for Ocean Research.”

Calibration and validation:
Need to state the location and spatial and temporal resolution of the data used for validation.  

Response: Text has been added to respond this comment. (Page 8, lines 263-268)

“The hourly water surface elevation was observed at the Danshui River mouth, Tu-Di-Gong-Bi, Taipei Bridge, Hsin-Hai Bridge, Zhong-Zheng Bridge, Bailing Bridge, and Dazhi Bridge gauge stations. Every half-hour longitudinal velocity was measured at the Kuan-Du Bridge, Taipei Bridge, Hsin-Hai Bridge, Zhong-Zheng Bridge, and Bailing Bridge, while half-hourly salinity was measured at the Kuan-Du Bridge, Taipei Bridge, and Bailing Bridge.”

Discussion:
This is really a 'Results' section so change the name.
I think there needs to be a more quantified description of the development changes to the system. For example, overall, by how much has the estuary volume changed? And the widths and depths? This information will help readers interpret your results more generally.

Response: 1. The original Section “5. Discussion” has been changed to “5. Results”. (Page 17, line 363)

  1. The tidal prism has been calculated and described in the revised manuscript. (Page 25, lines 477-487)

To explore the influence of geomorphic-driven changes on tidal prism, different scenarios of freshwater discharges imposed at upstream boundaries were used to calculated tidal prism. The equation of Lakhan [50] was adopted to yield the tidal prism. The equation is expressed as,, where P expresses the tidal prism, H denotes the mean tidal range, and A represents the surface area. The results revealed that the tidal prism for Q10 high flow condition was 8.91x107 m3 under the predevelopment condition, decreasing to 6.45x107 m3 under present condition. The tidal prism for Q75 low flow condition under the predevelopment condition was 8.8x107 m3, decreasing to 6.40x107 m3 under present condition. We found that the tidal prism under present condition was less than that under predevelopment condition as a result of geomorphic-driven changes. In the channel, the water volume below the mean sea level under predevelopment condition was also larger than that under present condition.”

  1. The depths and widths under predevelopment and present conditions have been presented in Figure 2 and Figure 3.

L364: You ran the model under several flow conditions, but what would happen under more realistic hydrograph conditions, such as a peak flow event rather than constant flow?

Response: Text has been added to respond this comment in the revised manuscript. (Page 20, lines 414-418)

“The long-term historically daily freshwater discharges in three main tributaries were analyzed to yield the flow equal to or exceeding percentage of time. Even though the flows presented in Table 4 could not represent the realistic hydrograph conditions, they were obtained from the statistical analysis results to cover different scenarios.”

A separate discussions section should be presented that generalises the results and puts them into a general or global context.

Response: The “6. Discussion” section has been added and separated from “5. Results”. (Page 25, lines 476-518)

And assumptions and future recommendations in this field.

Response: Assumption and future recommendation are added in the revised manuscript. (Page 25, lines 510-518)

To analyze the impacts of geomorphic-driven changes on hydrodynamics, saltwater intrusion, and residence time in tidal estuarine system, we adopted the same value of bottom roughness height under predevelopment and present conditions to assume the same bottom friction because no observed data were used to calibrate and validate the model under predevelopment condition. This assumption would be produced some biases in model simulation under predevelopment condition. However we believed that the simulated results and patterns would not have much change. The wind-wave action is not included in current study. Future recommendation is to consider the wind-wave effect, especially when the sediment transport is included in the model [59]. “

Round 2

Reviewer 2 Report

Seems thay have addressed the majority of our points so happy to accept.